METHODS AND RESOURCES

# Single-cell and spatiotemporal profile of ovulation in the mouse ovary

Ruixu Huang[1☯], Caroline E. Kratka[2☯], Jeffrey Pea[2], Cai McCann[3], Jack Nelson[3], John P. Bryan[3,9], Luhan T. Zhou[2], Daniela D. Russo[4,5,6], Emily J. Zaniker-Gomez[2], Achla H. Gandhi[1], Alex K. Shalek[4,5,6], Brian Cleary[7], Samouil L. Farhi[3‡], Francesca E. Duncan[2‡*], Brittany A. Goods [1,8‡*]

1 Thayer School of Engineering at Dartmouth College, Hanover, New Hampshire, United States of America, 2 Department of Obstetrics and Gynecology, Feinberg School of Medicine, Northwestern University, Chicago, Illinois, United States of America, 3 Spatial Technology Platform, Broad Institute of MIT and Harvard, Cambridge, Massachusetts, United States of America, 4 Klarman Cell Observatory, Broad Institute of MIT and Harvard, Cambridge, Massachusetts, United States of America, 5 Ragon Institute of MGH, MIT and Harvard, Cambridge, Massachusetts, United States of America, 6 Institute for Medical Engineering & Science, Department of Chemistry, and Koch Institute for Integrative Cancer Research, Massachusetts Institute of Technology, Cambridge, Massachusetts, United States of America, 7 Faculty of Computing and Data Sciences, Department of Biomedical Engineering, Department of Biology, Program in Bioinformatics, Boston University, Boston, Massachusetts, United States of America, 8 Molecular and Systems Biology, and Program in Quantitative Biomedical Sciences at Dartmouth College, Hanover, New Hampshire, United States of America, 9 Department of Electrical Engineering and Computer Science, Massachusetts Institute of Technology, Cambridge, Massachusetts, United States of America

☯ These authors contributed equally to this work.
‡ SLF, FED, and BAG also contributed equally to this work.
* f-duncan@northwestern.edu (FED); britt.anne.goods@dartmouth.edu (BAG)

## Abstract

Ovulation is a spatiotemporally coordinated process that involves several tightly controlled events, including oocyte meiotic maturation, cumulus expansion, follicle wall rupture and repair, and ovarian stroma remodeling. To date, no studies have detailed the precise window of ovulation at single-cell resolution. Here, we performed parallel single-cell RNA-seq and spatial transcriptomics on paired mouse ovaries across an ovulation time course to map the spatiotemporal profile of ovarian cell types. We show that major ovarian cell types exhibit time-dependent transcriptional states enriched for distinct functions and have specific localization profiles within the ovary. We also identified gene markers for ovulation-dependent cell states and validated these using orthogonal methods. Finally, we performed cell–cell interaction analyses to identify ligand-receptor pairs that may drive ovulation, revealing previously unappreciated interactions. Taken together, our data provides a rich and comprehensive resource of murine ovulation that can be mined for discovery by the scientific community.

**Data availability statement:** The single-cell transcriptomic data generated in this study are available at the Gene Expression Omnibus (GEO) under accession number GSE294534. The spatial transcriptomic data is available at Zenodo: https://doi.org/10.5281/zeno-do.15306735. All code used for analysis is publicly available at Zenodo: https://doi.org/10.5281/zenodo.15239554.

**Funding:** This work is supported by the Bill & Melinda Gates Foundation (INV-00385 to A.K.S., F.E.D., and B.A.G., https://www.gatesfoundation.org) and the National Institutes of Health (U01 MH130962 and RF1 MH121289 to B.C. and S.L.F., https://www.nih.gov). Under the grant conditions of the Bill & Melinda Gates Foundation, a Creative Commons Attribution 4.0 Generic License has been already assigned to the Author Accepted Manuscript version that might arise from this submission. C.E.K. is supported in part by the Eunice Kennedy Shriver National Institute of Child Health and Human Development of the National Institutes of Health under Award Number T32HD094699 (https://www.nichd.nih.gov). B.A.G. is supported in part by the Geisel School of Medicine at Dartmouth's Center for Quantitative Biology through a grant from the National Institute of General Medical Sciences (NIGMS, P20GM130454) of the NIH (https://www.nigms.nih.gov). We would also like to acknowledge NIH Grant 1S10OD025120 for the 10x Chromium housed in the NUSeq Facility. The content is solely the responsibility of the authors and does not necessarily represent the official views of the National Institutes of Health. None of the funders listed above were involved in the study design, data collection and analysis, decision to publish, or preparation of the manuscript.

**Competing interests:** I have read the journal's policy and the authors of this manuscript have the following competing interests: A.K.S reports compensation for consulting and/or SAB membership from Honeycomb Biotechnologies, Cellarity, Ochre Bio, Relation Therapeutics, Bio-Rad Laboratories, Passkey Therapeutics, Fog Pharma, Dahlia Biosciences, and intrECate Biotherapeutics. B.A.G., R.H., C.E.K., Y.Z., S.L.F., C.M., A.K.S., F.E.D., H.C.L., L.T.Z., E.Z., and J.N. have filed a patent related to the work described in this study (patent application no. PCT/US24/25824).

**Abbreviations :** CCI, cell–cell interaction; COC,

## Introduction

Ovulation is a highly dynamic and cyclic process that occurs approximately 650 and 400 times throughout the reproductive life span in mice and humans, respectively [1]. During ovulation, there is substantial structural remodeling of the ovary, where the antral follicle, comprised of an oocyte and surrounding supporting somatic cells, will release a mature egg for fertilization. A surge of luteinizing hormone (LH) initiates ovulation and orchestrates key events in the ovulatory cascade, including the resumption of meiosis in the oocyte, expansion of the cumulus layer of the cumulus-oocyte-complex (COC), follicular rupture, COC release, and luteinization of the residual follicular cells [2]. The intricate spatiotemporal and molecular changes in gene expression that accompany the extraordinary structural changes in the ovary during ovulation, which occur over a narrow window of hours, remain to be fully elucidated. Understanding the molecular drivers of ovulation has critical implications for the diagnosis and treatment of infertility, the discovery of novel contraceptives, elucidating species differences in reproduction, unraveling novel cell types that promote healthy ovulation, and uncovering possible mechanisms of ovarian aging [3–5].

Single-cell transcriptomic approaches have significantly advanced our ability to unravel the complexities of cell populations within tissues. Single-cell RNA sequencing (scRNA-seq) allows for the identification of molecular signatures defining specific cell states, the inference of communication patterns between cells, and the identification of genes and pathways that drive specific biological processes [6–11]. Single-cell imaging spatial transcriptomics (iST), meanwhile, is particularly well-suited for defining the spatial organization of gene expression within tissues at single-cell resolution, providing valuable insights into the spatial relationships between cells within the tissue microenvironments and limiting artifacts that can be introduced during dissociation [12]. As such, the combination of scRNA-seq and iST has been adopted to study tissue dysfunction, diseases, and treatment responses, and when combined with novel computational methods, can uncover deep insights into complex cellular biology [7,13]. In the context of the ovary, single-cell studies have been instrumental in demarcating genes that define certain cell types, elucidating signals that drive follicle activation, and discovering underlying mechanisms of diseases such as ovarian cancer [14–21]. These efforts have provided foundational insights into the ovary but have two main limitations. First, they have focused on the longer time-scale changes that occur across the whole estrous cycle, or second, they have relied on inferred single-cell genomic data rather than direct single-cell profiling [19,22]. Thus, there is a critical need to build a resource that comprehensively defines ovulation specifically and spatiotemporally at true single-cell resolution.

To address this, we conducted scRNA-seq and iST in parallel on paired mouse ovaries across three well-defined ovulatory timepoints. We then integrated these two modalities to generate a robust dataset comprised of 391,584 cells via iST and 26,411 cells via scRNA-seq, capturing dynamic variations in gene expression and corresponding morphological changes throughout the ovulatory process in mice. Our analysis reveals major cell types (cumulus, stromal, granulosa, theca, luteal) with associated time-dependent gene signatures and localization within the ovary across

cumulus-oocyte-complex; DEA, differential gene expression analysis; ETC, electron transport chain; GEO, gene expression omnibus; GO, gene ontology; hCG, human chorionic gonadotropin; iST, imaging spatial transcriptomics; LH, luteinizing hormone; LUF, luteinized unruptured follicle; MMPs, matrix metalloproteinases; OCT, optimal cutting temperature; SDS, sodium dodecyl sulfate; SSC, saline–sodium citrate; TIMPs, tissue inhibitor metalloproteinases; TMA, tissue-microarray.

ovulation. We also identify previously unappreciated genes and pathways within these ovulation-dependent cell clusters, such as neuronal pathways in cumulus cells and metal homeostasis in theca cells. Finally, we conducted cell–cell interaction (CCI) analyses to discover ligand-receptor pairs that may drive cell state changes in cumulus cells and underpin the luteinization process in granulosa and theca cells. Taken together, our resource provides a comprehensive view of ovulation, supporting several critical avenues of research related to reproductive health.

## Results

### A single-cell temporal reference of cell types in the adult mouse ovary across ovulation reveals "early" and "late" cell states

To deeply profile the mouse ovary across ovulation using scRNA-seq and iST, we induced ovulation with human chorionic gonadotropin (hCG) in a synchronized fashion following hyperstimulation, and ovaries were collected at 0 hr, 4 hr, or 12 hr after induction (Fig 1A). The 4 hr time point is early in the ovulation process and represents the peak of the LH surge when the expression of key ovulation regulators is highest; the 12 hr time point, meanwhile, represents a later stage when follicular rupture is underway [23]. We confirmed this with histological analysis of ovaries collected from a parallel set of mice (S1 Fig). At 0 hr post-hCG, there were follicles at different stages of development present including numerous antral follicles which were recruited to grow following hyperstimulation. Corpora lutea from previous estrous cycles were also present (S1A Fig). By 4 hr post-hCG, the response to the ovulatory cue was apparent given the increase in the size of the antral follicles due to the accumulation of follicular fluid and expansion of the antral cavity and the cumulus layer surrounding the COCs (S1B Fig). By 12 hr post-hCG, there were few antral follicles left in the ovary suggesting that ovulation had taken place, and consistent with this, the corpora lutea were prominent at this time point (S1C Fig). The few follicles that remained in the ovary at this time were close to rupture given their large size, thinning of the mural granulosa layer close to the ovarian surface, and the presence of fully expanded COCs containing mature metaphase II arrested eggs (S1C Fig). We collected one ovary per mouse for scRNA-seq and the contralateral ovary for iST at each of these specific timepoints (Figs 1A and S2).

For the generation of scRNA-seq data, we followed standard 10X genomics workflows post-dissociation of ovarian tissues (see "Materials and methods"). This resulted in a high-quality data set comprised of 26,411 cells (S3A–3C Fig). We then used standard computational workflow to identify major top-level ovarian cell identities across ovulation and their associated marker genes [24] (see "Materials and methods"). We identified eight major cell types in the ovary: cumulus cells, endothelial cells, epithelial cells, granulosa cells, luteal cells, myeloid cells, stromal cells, and theca cells (Fig 1B). Additionally, a cluster of oocytes (n = 16 cells) was identified but not further analyzed because of the small cluster size. The major cell types were characterized by highly enriched marker gene expression (Fig 1C and Table 1). When compared to an existing scRNA-seq reference of the ovary generated throughout the estrous cycle, we successfully recapitulated all major ovarian cell

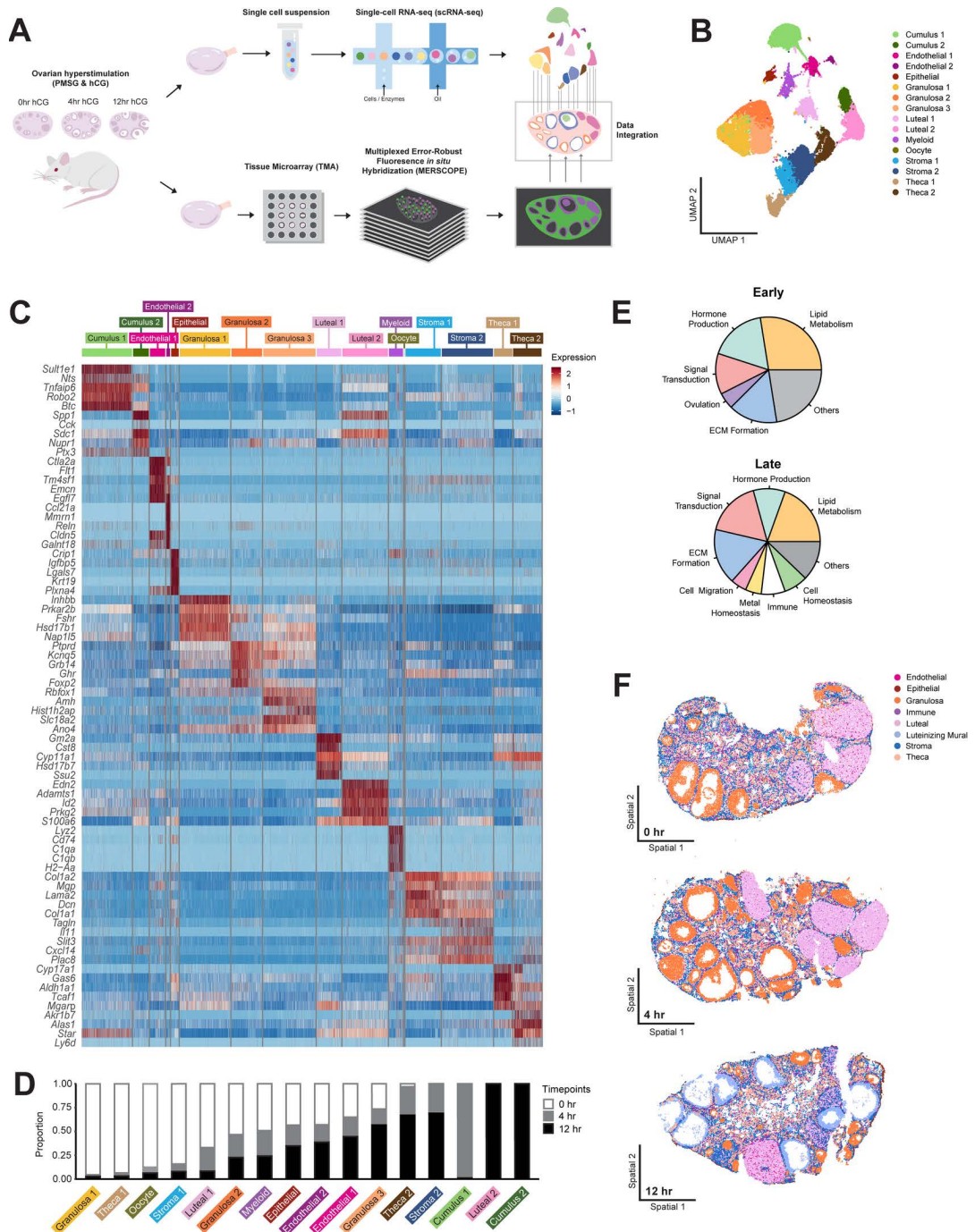

**Fig 1. Single-cell and spatial transcriptomic analysis of adult mouse ovaries throughout the time course of ovulation. (A)** Schematic depicts the workflow for single cell and spatial transcriptomic analyses. Mice were hyperstimulated with PMSG and hCG and ovaries were dissected 0 hr, 4 hr, or 12 hr post-hCG injection, processed, and submitted for scRNA-seq or MERFISH analysis. **(B)** UMAP shows all 16 identified cell type clusters. The cell counts for each cluster were as follows: Cumulus 1 = 2,910, Cumulus 2 = 908, Endothelial 1 = 891, Endothelial 2 = 212, Epithelial = 440, Granulosa 1 = 2,931, Granulosa 2 = 1,781, Granulosa 3 = 3,044, Luteal 1 = 1,421, Luteal 2 = 2,641, Myeloid = 816, Oocyte = 16, Stroma 1 = 2,042, Stroma 2 = 2,970, Theca 1 = 1,051, Theca 2 = 1,640. **(C)** Heatmap shows five marker genes used to determine the identity of each cell cluster. **(D)** Stacked bar plot showing the percent of cells in each cluster expressed at each time point. **(E)** Pie charts showing the categorization of pathways upregulated in early (0 hr) vs. late (12 hr) timepoints. **(F)** Examples of 0 hr, 4 hr, and 12 hr MERFISH ovary sections with seven major cell types localized. The remaining ovaries can be visualized in S4B Fig. The data underlying this figure is available at the Gene Expression Omnibus (GEO) under accession number GSE294534.

types [17] (S3E Fig). Furthermore, within each major cell type, we identified unique cell subclusters for cumulus (Cumulus 1 and 2), endothelial (Endothelial 1 and 2), granulosa (Granulosa 1, 2, and 3), luteal (Luteal 1 and 2), stroma (Stroma 1 and 2), and theca (Theca 1 and 2) cells (Fig 1B). Several known marker genes were enriched in these particular clusters, including *Sult1e1* (cumulus), *Inhbb* and *Amh* (granulosa), *Lhcgr* (luteal), *Dcn* (stroma), and *Cyp17a1* (theca). We also identified previously undescribed highly expressed genes specifically found in certain clusters, including *Zpf804a* (cumulus), *Oca2* (theca), *Pdzrn3* (stroma), and *Gm2a* (luteal) (Fig 1C and Table 1). Notably, we found two cumulus cell clusters that were unique to ovulation given their temporal emergence, which were identified based on cumulus cell expansion markers and gene ontology (GO) pathway analysis results. These clusters of expanding cumulus cells are not present in a reference scRNA-seq dataset of the adult mouse ovary captured throughout the estrous cycle but are not limited to proestrus when ovulation takes place. The presence of these clusters only in our dataset demonstrates their unique relevance to ovulation (S3E Fig). By focusing our temporal data collection on the short window of ovulation, our dataset has a high enough resolution to detect cumulus cell populations that are undergoing unique processes and would otherwise not be present [17].

In addition to differences in gene expression, some cell clusters separated into "early" and "late" groups relative to the ovulation time course, while other clusters were present throughout all ovulation (Fig 1D). Specifically, "early" clusters included Theca 1, Granulosa 1, Stroma 1, and Luteal 1, which were expressed almost exclusively at 0 hr and/or 4 hr. "Late" clusters were expressed predominantly at 12 hr and included Cumulus 2, Luteal 2, Stroma 2, and Theca 2. Clusters expressed throughout ovulation included Granulosa 2, Granulosa 3, Endothelial 1, Endothelial 2, Epithelial, and Myeloid. To determine if there were underlying functional differences across clusters, we performed pathway analysis on the marker gene list for each identified cluster of cells and categorized the pathways by high-level functional classifications (see "Materials and methods", S1 File). We then compared the representation of these high-level functions across early or late clusters (Fig 1E). Pathways related to "signal transduction", "hormone production," "ECM formation", and "lipid metabolism" had similar representation in the early and late clusters. However, there were several distinct pathway categories in early and late clusters. This included "ovulation" (*Inha*, *Inhba*, *Inhbb*, *Foxl2*) in the early clusters, and pathways related to "cell migration" (*Col3a1*, *Timp1*, *Fgf2*), "cell homeostasis" (*Col1a1*, *Col1a2, Sparc*), "metal homeostasis" (*Mt1*, *Mt2*, *Mt3*) and "immune function" (*Hspa8, Cd63, Anxa2, Gstp1*) in the later clusters (S1 File). Taken together, these results demonstrate that early and late cell clusters can be broadly distinguished by shifts in their functional pathways, where there is a greater diversity of pathway terms in the clusters associated with the later time point in ovulation.

**Table 1. Marker genes for top-level clusters.**

| | Cluster 1 | Cluster 2 | Cluster 3 | Cluster 4 | Cluster 5 | Cluster 6 | Cluster 7 | Cluster 8 | Cluster 9 | Cluster 10 | Cluster 11 | Cluster 12 | Cluster 13 | Cluster 14 | Cluster 15 |
|---|---|---|---|---|---|---|---|---|---|---|---|---|---|---|---|
| | *Cumulus 1* | *Cumulus 2* | *Endothelial 1* | *Endothelial 2* | *Epithelial* | *Granulosa 1* | *Granulosa 2* | *Granulosa 3* | *Luteal 1* | *Luteal 2* | *Myeloid* | *Stroma 1* | *Stroma 2* | *Theca 1* | *Theca 2* |
| 1 | Sult1e1 | Spp1 | Ctla2a | Ccl21a | Crip1 | Inhbb | Ptprd | Rbfox1 | Gm2a | Edn2 | Lyz2 | Col1a2 | Tagln | Cyp17a1 | Akr1b7 |
| 2 | Nts | Sdc1 | Flt1 | Mmrn1 | Igfbp5 | Prkar2b | Kcnq5 | Amh | Cst8 | Adamts1 | Cd74 | Mgp | Il11 | Gas6 | Alas1 |
| 3 | Tnfaip6 | Nupr1 | Tm4sf1 | Reln | Lgals7 | Fshr | Grb14 | Hist1h2ap | Cyp11a1 | Id2 | Apoe | Lama2 | Slit3 | Aldh1a1 | Cyp11a1 |
| 4 | Robo2 | Ptx3 | Emcn | Gng11 | Krt19 | Hsd17b1 | Ghr | Slc18a2 | Hsd17b7 | Prkg2 | C1qa | Dcn | Cxcl14 | Tcaf1 | Star |
| 5 | Btc | Lox | Egfl7 | Cldn5 | Plxna4 | Nap1l5 | Foxp2 | Ano4 | Ssu2 | S100a6 | C1qb | Col1a1 | Plac8 | Mgarp | Ly6d |
| 6 | Pgr | Cited4 | Igfbp7 | Galnt18 | Lgals2 | Inhba | Sema5a | Tanc2 | Akr1c18 | Psap | H2-Aa | Ogn | Pdzrn3 | Fabp3 | Hao2 |
| 7 | Areg | Rgcc | Ramp2 | Prox1 | Lgals1 | Grem2 | Foxo1 | Hist1h2ae | Lhcgr | Rgcc | H2-Ab1 | Igfbp7 | Cald1 | Dnajc15 | Rhox8 |
| 8 | Tac1 | Lgals3 | Eng | Aqp1 | Krt18 | Tnni3 | Zfp385b | Fam13a | Ptgfr | Ephx2 | Tyrobp | Col3a1 | Lhfp | Pank1 | Abca1 |
| 9 | Alcam | Pik3c2g | Ehd4 | Lyve1 | Upk3b | Inha | Pde7b | Mctp1 | Idh1 | Mt2 | Fcer1g | Acta2 | Abi1 | Nckap5 | Cyp51 |
| 10 | Ptgs2 | Ier3 | Ptprb | Cavin2 | Epcam | Fam13a | Tenm4 | Hist1h1b | Sfrp4 | Slco2a1 | C1qc | Cped1 | Tenm3 | Akr1cl | Me1 |

## Single-cell spatial transcriptomic analysis of adult mouse ovaries throughout ovulation reveals temporally-driven processes

To identify genes and signaling interactions driving spatiotemporal changes across ovulation, we also generated single-molecule resolution iST data on collected contralateral ovaries (Figs 1A and S2C–2D). We designed a 198-plex MER-SCOPE panel based on known marker genes for ovarian cell types and genes of interest relevant to ovulation from previously published datasets (S2 File) [17,18,23]. We assembled contralateral ovaries into an optimal cutting temperature (OCT) tissue microarray to allow parallel processing of all ovaries on a single coverslip and thus minimize batch effects (S2C Fig). We acquired data from 8 samples (3 ovaries at 0 hr, 3 ovaries at 4 hr, and 2 ovaries at 12 hr). As expected, the anatomical features of the MERSCOPE sections at 0 hr and 4 hr post-hCG were very similar to those observed via histology (Figs 1F, S1A–1B and S4B). At 12 hr post-hCG, one MERSCOPE section was from an ovary containing mostly corpora lutea and few unruptured follicles (S4B Fig) similar to what we observed via histology (S1C Fig), whereas the other MERSCOPE section was from an ovary with many peri-ovulatory follicles close to rupture (Fig 1F). The presence of these two phenotypes at 12 hr post-hCG is consistent with inherent biological variation in the timing of ovulation.

Following filtering based on transcript count and cell area, we retained 391,584 cells across all samples, with an average of 88 transcripts per cell (S3C–3D Fig). We then normalized and scaled counts, performed dimensionality reduction and Leiden clustered, and annotated cell types in each ovary based on genes enriched in each cluster. We additionally used ovarian morphology to annotate the major top-level cell types, including theca cells, stromal cells, granulosa cells, luteal cells, immune cells, endothelial cells, and epithelial cells (Figs 1F and S4A–4C). To further investigate spatiotemporal gene expression changes during ovulation, we integrated our single-cell and spatial transcriptomes together with a machine learning approach [25]. Our integrated data resource powers us to precisely identify high-resolution spatiotemporal changes in genes beyond those initially targeted by our MERSCOPE probes. As a result, we could infer the localization patterns of over 25,000 genes from ovaries at 0 hrs, 4 hrs, and 12 hrs post-hCG. We evaluated the integration with multiple approaches (see "Materials and methods"). First, more than 70% of genes having a training score above 0.7 demonstrates the successful integration at the model level (S5A Fig). Second, we randomly selected test genes and excluded them from the model training process. When comparing the predicted expression profile versus the actual expression profile from the spatial transcriptome, we observe similar localization patterns for those test genes such as *Col1a2*, which is highly enriched in mesenchymal cells (S5B Fig). Lastly, we compared predicted gene expression profiles with actual RNA expression detected using RNAscope on histological ovarian sections as validation, where the predicted expression profile of *Sik3* was primarily present in luteal cells at 0 hr and *Slc6a6* being primary present in cumulus and mural granulosa cells at 12 hr was recapitulated in RNAscope images (S5C Fig).

Through scRNA-seq and iST analyses, we have captured all major ovarian cell types and their spatial distributions across ovulation which are consistent with established timelines of murine ovulation and expected changes in follicular morphology. To date, this is the most comprehensive single-cell dataset describing the events of ovulation in the mouse and serves as a robust resource for discovering cell states that are unique to ovulation.

## Analysis of time-dependent cell types reveals unique expression patterns and putative functions of early and late cell clusters

To further investigate the time-dependent changes in gene expression for each cell type, we performed sub-clustering analyses where possible, followed by pathway analyses on cluster-defining marker genes and differential gene expression analysis (DEA) between early and late cell subclusters for time-varying cell types.

**Stromal cells.** We observed time-dependent changes in gene expression throughout ovulation and identified two stromal cell subclusters: early stromal (Stroma 1) and late stromal (Stroma 2) cells (Fig 2A and 2B). DEA performed between the early and late stromal subclusters indicated that the top-upregulated genes in early stromal cells are *Lama2*, *Grm7*, *Ptprd*, *Tcf21*, and *Tenm4*, whereas the top-upregulated genes in late stromal cells are *Timp1*, *Ereg*, *Il11*, *Pde10a*,

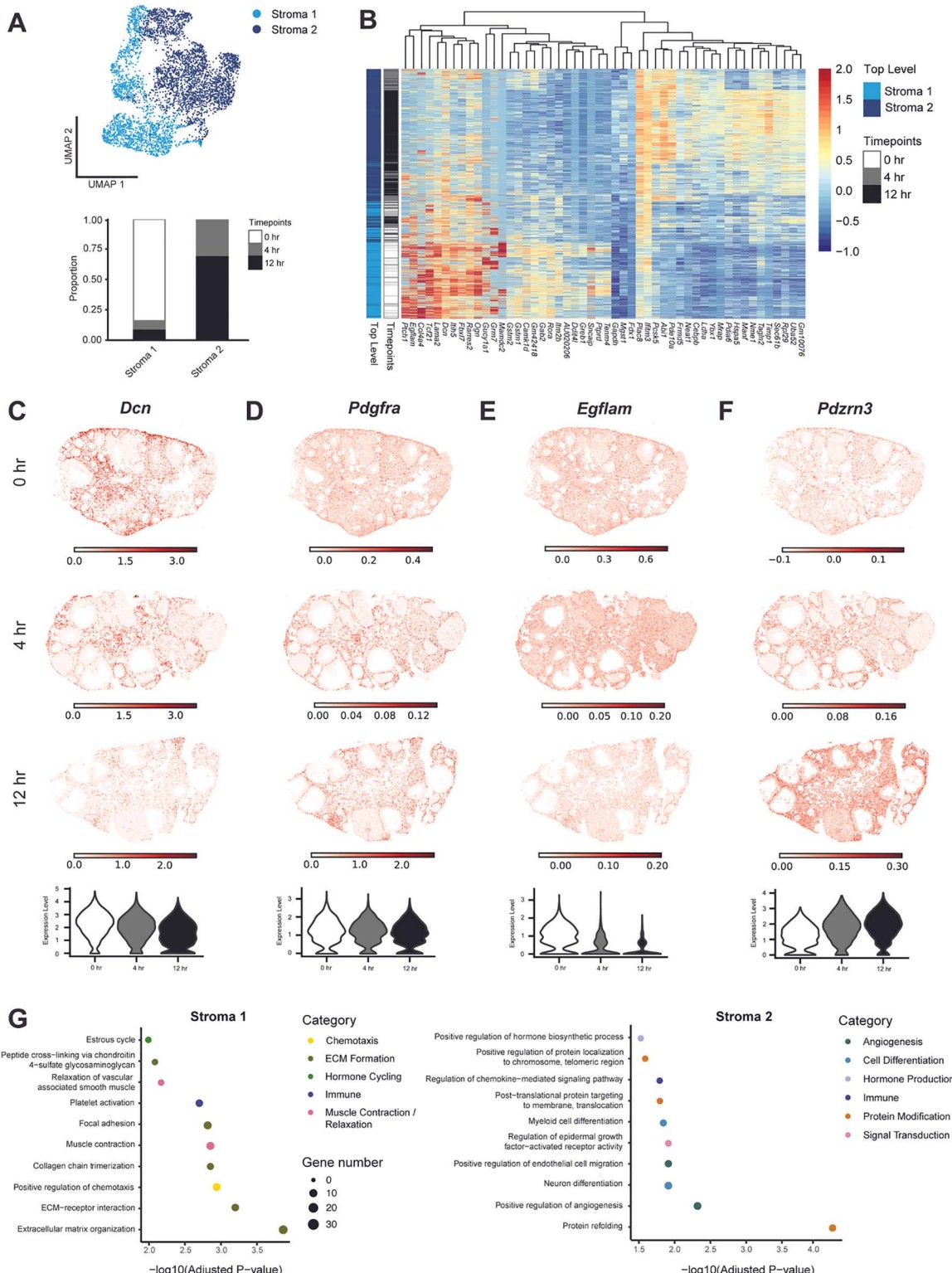

**Fig 2. Stroma cells exhibit time-dependent changes in gene expression. (A)** UMAP shows the clustering of stroma cells from scRNA-seq data. A stacked bar plot shows the percent of cells in stromal clusters expressed at each time point. **(B)** Heatmap depicts differential gene expression in Stroma 1 (early) and Stroma 2 (late) clusters from scRNA-seq data. **(C)** Inferred expression of Dcn from the integration of scRNA-seq and iST data shows

localization to stroma cells. **(D)** Expression plot of Pdgfra from iST data shows localization to stroma cells. **(E)** Same with C but with *Egflam.* **(F)** Same with C but with *Pdzrn3.* **(G)** Dot plots show top processes upregulated in Stroma 1 (left) and Stroma 2 (right). The data underlying this figure is available at the Gene Expression Omnibus (GEO) under accession number GSE294534.

and *Mrap* (Fig 2B and S3 File). Within each subcluster, we further characterized the expression profiles of known ovarian stromal cell markers. *Dcn,* or decorin, is a proteoglycan associated with extracellular matrices in a variety of tissues and was highly enriched in early stromal cells [26] (Fig 2C). In addition, *Pdgfra*, a platelet-derived growth factor, and ovarian mesoderm marker, exhibited increased expression in late stromal cells [27,28] (Fig 2D). We also identified genes that were highly enriched but not well characterized in the ovarian stroma, such as *Egflam* (early stromal) and *Pzdrn3* (late stromal). *Egflam*, also known as pikachurin, is an extracellular matrix-like protein that interacts with dystroglycan and has been implicated in photoreceptor synapse function [29,30] (Fig 2E). *Pdzrn3* encodes an E3 ubiquitin ligase previously shown to be involved in cell migration and vascular morphogenesis [31,32] (Fig 2F).

GO Analysis of top differentially expressed genes between the early stromal (Stroma 1) and late stromal (Stroma 2) subclusters revealed that early stromal cells were enriched in pathways related to ECM formation (*Fbn2*, *Lama2*, *Lum*), smooth muscle relaxation (*Gucy1a1*, *Prkg1*, *Mrvi1*), and the estrous cycle (*Mdk*, *Ptn*, *Anxa1*). Late stroma cells were unique in pathways related to angiogenesis (*Xbp1*, *Pkm*, *Bmper*), cell differentiation (*Runx1*, *Ugdh*, *Hspa9*), hormone production (*Wnt4*, *Hif1a*, *Por*), and protein modification (*Hsp90aa1*, *Hspa9, Hspd1*) (Fig 2G and S3 File). The upregulation of ECM-related pathways in early stroma cells suggests that this population has greater involvement in general ECM organization and remodeling than late stroma cells. The enrichment of hormone production pathways in late stromal cells suggest functions of late stromal cells in steroid hormone production. In addition, late stroma cells potentially increase blood vessel formation to accommodate the transport of these hormones to the rest of the body.

**Theca cells.** Theca cells were further differentiated into four subclusters, each detected at one specific time point (0 hr, 4 hr, or 12 hr); two subclusters were predominantly observed at 0 hr whereas the remaining two subclusters were found at 4 hr or 12 hr (Fig 3A–3B and S4 File). The top differentially expressed genes in the theca subclusters are shown in Fig 3B. Genes known to be active in theca cells include *Cyp17a1* and *Col4a1*, which are expressed in early and late theca cells, respectively. More specifically, *Cyp17a1* expression was distinguished at 0 hr from the 4 hr and 12 hr theca subclusters, which were marked by *Col4a1* expression (Fig 3C and 3D). *Cyp17a1* is a cytochrome P450 enzyme that catalyzes critical steps in androgen synthesis [33–35], whereas *Col4a1* is an ECM component that is part of the theca basement membrane [36,37]. In addition, we identified several genes that were upregulated in theca subclusters that have yet to be described in this cell type, including *Oca2* (at 0 hr) and *Fam161a* (at 12 hr) (Fig 3E and 3F). *Oca2* encodes P protein, which is integrated into the cell membrane and functions by maintaining pH and transporting small molecules [38]. In contrast, *Fam161a* encodes for a centrosomal protein primarily known for ciliary function and microtubules [39–41].

We then conducted GO analysis of top differentially expressed genes between each theca subcluster (Fig 3G). This analysis revealed that in the first 0 hr cluster (Theca_0 hA), pathways upregulated in theca cells included hormone production, immune function, and metabolism of lipids/phosphorus, which are consistent with the functions of a steroidogenic theca cell population. In the second 0 hr subcluster (Theca_0 hB), theca cells presented enrichment in processes related to hormone production, ovulation, and signal transduction. Theca cells at 4 hr showed upregulated pathways including chemotaxis and signal transduction whereas pathways upregulated in the 12 hr subcluster were related to cell proliferation, immune processes, metal homeostasis, and plasma membrane function (Fig 3G). Theca cells at 12 hrs up-regulate metal homeostasis pathways, driven by the genes *Mt1*, *Mt2*, and *Mt3*, which suggests that metal homeostasis could be important for theca cells to perform their primary functions at later timepoints in ovulation, including hormone synthesis. Taken together, the results of our GO analysis show that theca cell subclusters are enriched for known functions, like hormone production, as well as unexplored pathways performed by theca cells temporally.

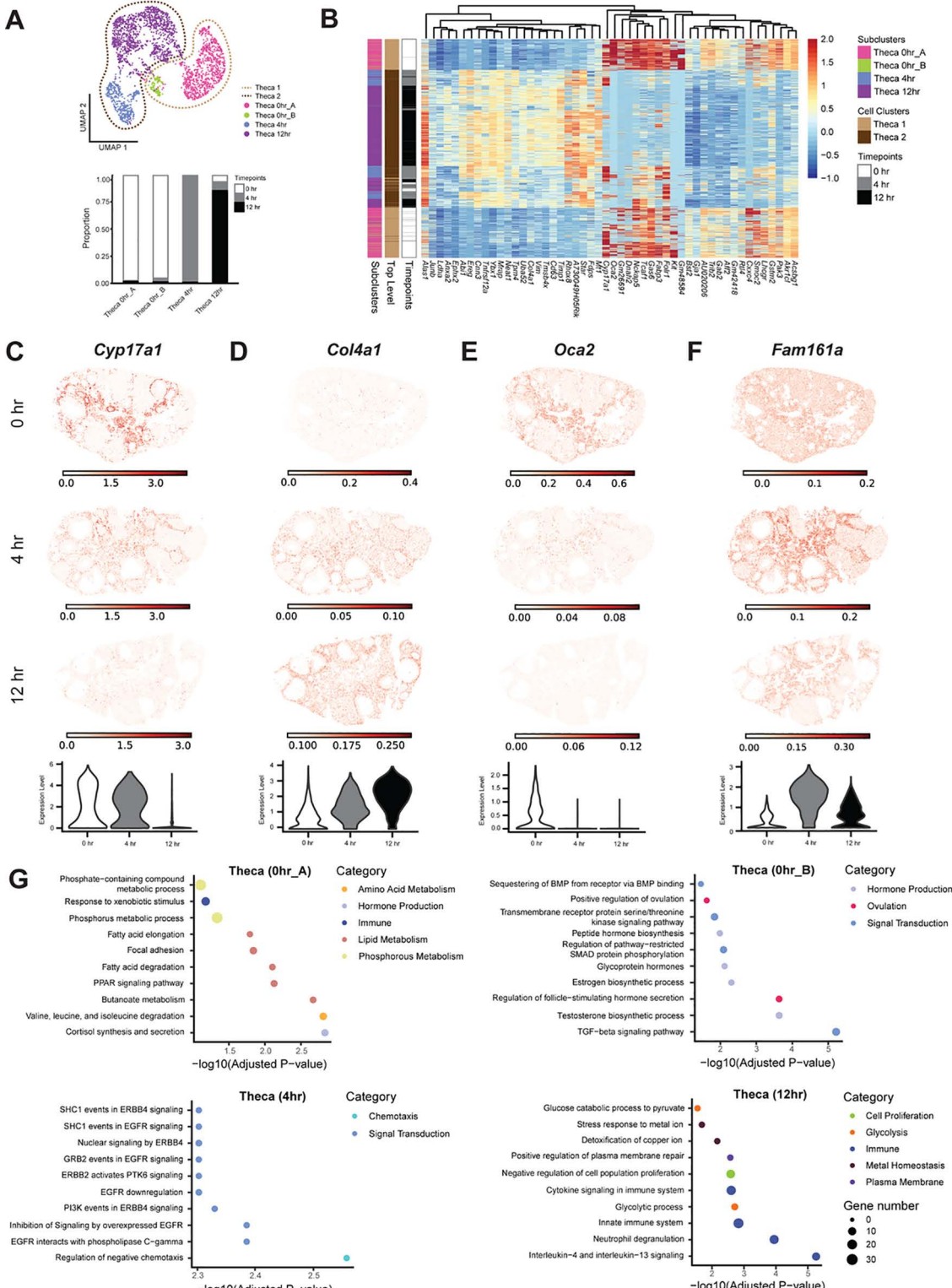

**Fig 3. Theca cells exhibit time-dependent changes in gene expression. (A)** UMAP shows the clustering of theca cells at the top-level (dotted line) and the subcluster level (dots) from scRNA-seq data. A stacked bar plot shows the percent in theca subclusters expressed at each time point. **(B)** Heatmap depicts differential gene expression in Theca 1 (early) and Theca 2 (late) clusters from scRNA-seq data. **(C)** Expression plot of *Cyp17a1* from

iST data shows localization to theca cells. **(D)** Inferred expression of *Col4a1* from the integration of scRNA-seq and iST data shows localization to theca cells. **(E)** Same with D but with *Oca2.* **(F)** Same with D but with *Fam161a.* **(G)** Dot plots showing top processes upregulated in Theca 0 hr_1 (top left), Theca 0 hr_2 (top right), Theca 4 hr (bottom left), and Theca 12 hr (bottom right). The data underlying this figure is available at the Gene Expression Omnibus (GEO) under accession number GSE294534.

**Luteal cells.**  Using published luteal cell subtype markers, we identified four luteal subclusters: Recent Luteal cells, Mural cells, Mixed Luteal cells, and Other Luteal cells (Fig 4A–4B and S4 File). Recent Luteal cells were defined by high expression of proliferation markers and steroidogenic enzymes and represent corpora lutea that are currently producing and secreting progesterone [17]. Mural cells were comprised of granulosa and theca cells of large antral follicles that are differentiating into luteal cells and were marked by expression of genes including *Adamts1* [42,43], *Mt2* [44–46], *Mrap* [47], *Parm1* [48], *S100a6* [49], and *Cdkn1a* [17]. Other Luteal cells were characterized by expression of the proliferation marker *Top2a*, as well as other marker genes including *Ube2c* [50], *Birc5* [17], *Inhbb* [51], and *Nap1l5* [52]. Mixed Luteal cells expressed various luteal markers and could not be distinguished into a specific luteal cell subtype, but likely consist of a mix of active and regressing corpora lutea with varying levels of progesterone secretion. The top differentially expressed genes in all subclusters are shown in Fig 4B. The luteal subclusters exhibited temporal differences with Mixed Luteal cells predominantly present at the 0 hr and 4 hr timepoints, Other Luteal cells expressed at 0 hr and 12 hr, and Recent Luteal and Mural Luteal cells primarily found at 12 hr (Fig 4B).

We further investigated the differential gene expression profiles between all subclusters. As expected, we observed upregulation of *Lhcgr* in Mixed Luteal and Other Luteal cells (Fig 4C). LH binds to LHCGR to promote androgen synthesis. In addition, *Runx1*, a DNA-binding protein known to be an essential transcriptional regulator of ovulation and luteinization, is upregulated at 12 hr in the Recent Luteal and Mural Luteal subclusters [53–55] (Fig 4D). We also identified previously unappreciated genes with temporal expression patterns in luteal cells. *Gm2a*, a glycolipid transporter that promotes the degradation of ganglioside GM2 to GM3 enriched in the Mixed Luteal and Other Luteal subclusters [56] (Fig 4E). Notably, GM3 is upregulated in the rat ovary during ovulation, but its role is largely unknown [57,58]. In addition, *Fndc3b* was enriched in the Recent Luteal and Mural Luteal subclusters and belongs to the fibronectin type III domain containing a family of myokines and adipokines with general roles in migration, adhesion, and proliferation of cells (Fig 4F). FNDC3B has roles in bone and tumor development, but no documented functions in the ovary [59–61]. However, another member of this myokine/adipokine family, FNDC5 or irisin, regulates the steroid hormone production and secretion in various granulosa cell models [62–64], suggesting that *Fndc3b* may function in the production of progesterone within the corpus luteum. Given the relationship between theca, granulosa, and luteal cells during ovulation, we assessed the co-localization of these three ovarian cell types during early and late timepoints (S6 Fig). We observed that early luteal cells (*Gm2a* - Luteal 1) are primarily found in the corpus lutea from previous estrous cycles at 0 hr. In comparison, marker genes for early theca cells (*Cyp17a1* – Theca 1) are localized around preantral and antral follicles whereas markers for early granulosa cells (*Inhbb* – Granulosa 1) are present primarily within antral follicles and correspond with mural granulosa cells at 0 hr post-hCG (S6A and S6C Fig). In contrast, marker genes for late luteal cells (*Fndc3b* – Luteal 2) are primarily localized to mural granulosa cells of antral follicles at 12 hr post-hCG (corresponding with mural luteal cells). At the same time, marker genes for late theca cells (*Col4a1* – Theca 2) continue to show localization around antral follicles but are less specific given their additional expression in the stroma (S6B Fig). Additionally, marker genes for late granulosa cells (*Amh* – Granulosa 3) are present primarily in preantral follicles at 12 hr post-hCG (S6D Fig).

We next performed GO analysis on the top level Luteal 1 and 2 clusters using the marker genes for each cluster (S1 File). Early luteal (Luteal 1) cells are highly enriched in pathways related to hormone production, consistent with the role of luteal cells in synthesizing progesterone. Late luteal (Luteal 2) cells exhibit a range of upregulated pathways, including those involved in extracellular matrix organization and signal transduction. We also observed "response to estrogen" and "cell volume homeostasis" as two additional pathways of interest in the Luteal 2 cluster (S1 File). High levels of estrogen

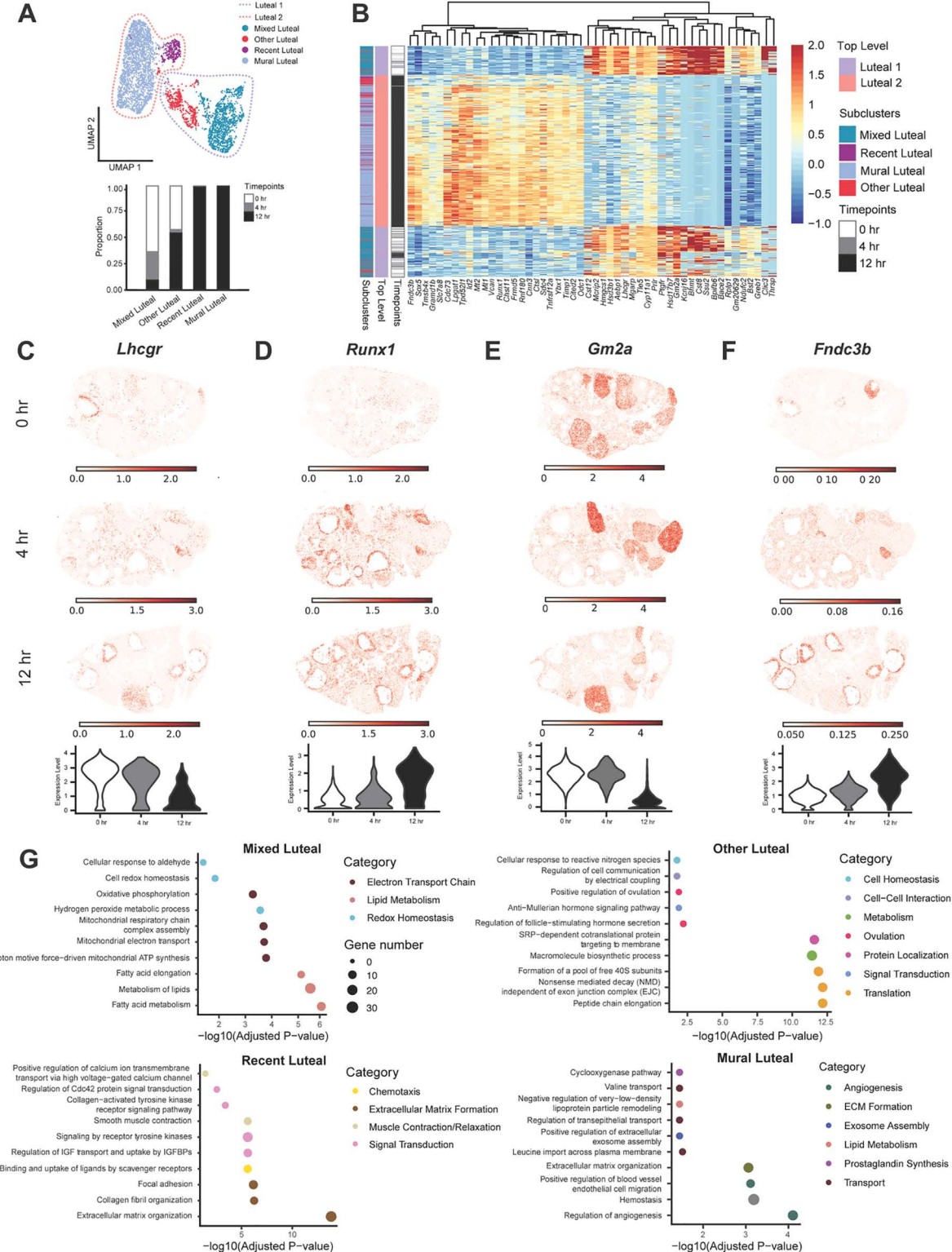

**Fig 4. Luteal cells exhibit time-dependent changes in gene expression. (A)** UMAP shows the clustering of luteal cells at the top-level (dotted line) and the subcluster level (dots) from scRNA-seq. A stacked bar plot shows the percent of cells in luteal subclusters expressed at each time point. **(B)** Heatmap depicts differential gene expression in Recent Luteal and Mixed Luteal subclusters from scRNA-seq. **(C)** Expression plot of *Lhcgr* from

iST data shows localization to luteal cells. **(D)** Same with C but with *Runx1*. **(E)** Same with C but with *Gm2a*. **(F)** Inferred expression of *Fndc3b* from the integration of scRNA-seq and iST data shows localization to stroma cells. **(G)** Dot plots show top processes upregulated in Mixed Luteal (top left), Other Luteal (top right), Recent Luteal (bottom right), and Mural Luteal (bottom left) subclusters. The data underlying this figure is available at the Gene Expression Omnibus (GEO) under accession number GSE294534.

trigger ovulation, which is followed by luteinization of granulosa and theca cells. In addition, luteinization is marked by hypertrophy of luteal cells – a modulation of cell volume. In interpreting these results, it is important to note that the Luteal 1 cluster (which contains cells from all three timepoints but mostly 0 hr and 4 hr) likely contains existing corpora lutea from previous cycles. In mice, corpora lutea persist for multiple cycles, receive repeated luteolytic signals, and may continue to produce progesterone [65,66]. On the contrary, the Luteal 2 cluster is exclusively present at 12 hr and likely represents actively forming or newly formed corpora lutea, supporting the presence of pathways related to the organization of luteinizing granulosa and theca cells into a functional corpus luteum.

Lastly, we conducted GO analysis on all luteal subclusters using the differentially expressed genes between each subcluster (Fig 4G). In Mixed Luteal cells, we observed several pathways related to lipid metabolism and redox homeostasis. In addition, genes that encode subunits of complex I of the electron transport chain (ETC), NADH:ubiquinone oxidoreductase (e.g., *Ndufa6, Ndufb4, Ndufab1, Ndufs3, Ndufc2, and Ndufc1*) drove upregulation of ETC-related pathways. Recent Luteal cells were enriched in pathways related to extracellular matrix formation, immune function, muscle contraction, and signal transduction. Other Luteal cells showed upregulation of pathways related to translation and localization of proteins. Mural Luteal cells were enriched in pathways including the metabolism of prostaglandins and lipids, and the formation of blood vessels and ECM (Fig 4G). Thus, these results suggest that Mixed Luteal cells exhibit a primary function of steroid hormone production, consistent with the synthesis of progesterone within mature corpus lutea from previous cycles. In contrast, Recent Luteal and Mural Luteal cells may represent newly forming corpus lutea that are undergoing extensive ECM remodeling and vascularization. Other Luteal cells appear to have active roles in protein synthesis.

**Cumulus cells.** For cumulus cells, we identified two time-dependent subclusters: early cumulus (Cumulus 1) and late cumulus (Cumulus 2) (Fig 5A and 5B). Both subclusters were distinct temporally, with early and cumulus cells primarily detected at 4 hr and 12 hr, respectively (Fig 5C). The top differentially expressed genes between the early and late cumulus cell subclusters were *Sult1e1*, *Robo2*, *Pgr*, *Rnf180*, and *Tac1* (upregulated in early cumulus cells), and *Spp1*, *S100a6*, *Cck*, *Timp1*, and *Chchd10* (upregulated in late cumulus cells; Fig 5B and S3 File). Given that the cumulus subclusters were not previously identified in the reference scRNA-seq dataset [17] (S3E Fig), we validated the single-cell expression pattern using in situ RNA hybridization on histological sections (Fig 5C–5F). Utilizing histological sections was advantageous for these experiments because the tissue sections utilized for MERFISH contained very few complete COCs for analysis, given that only one tissue section was used per ovary. In the histological samples, adjacent sections are only 5 μm apart and capture the entire COCs across sections, thus permitting selection of complete COC structures for target validation. In addition to the 0 hr, 4 hr, and 12 hr post-hCG timepoints, we also included an 11 hr time point to allow greater resolution of the late periovulatory period. As expected, *Sult1e1*, a gene known to be expressed in cumulus cells that are involved in estrogen metabolism, was high at 4 hr [67–69] (Fig 5C). Also as expected, *Lox,* a gene involved in ECM formation and expressed in developing follicles and COCs, was high at 12 hr [70–73] (Fig 5D). In addition, our transcriptomic analysis identified two genes, *Zfp804a* and *Emb*, which also showed similar temporal expression patterns to *Sult1e1* and *Lox*, respectively. We observed similar expression patterns with RNAscope validation (Fig 5E and 5F).

Gene ontology analysis of top marker genes in the Cumulus 1 and 2 clusters revealed "ovarian cumulus expansion" as a top biological process in both clusters, further validating our cell type identification (S1 File). Early cumulus cells also showed upregulation of "glycosaminoglycan metabolism", "fused antrum stage", and "EGFR interacts with phospholipase C-gamma" processes. These pathways are consistent with known biological processes occurring in the COC during ovulation, when EGF stimulates production of glycosaminoglycans (e.g., hyaluronan) in cumulus cells to promote COC

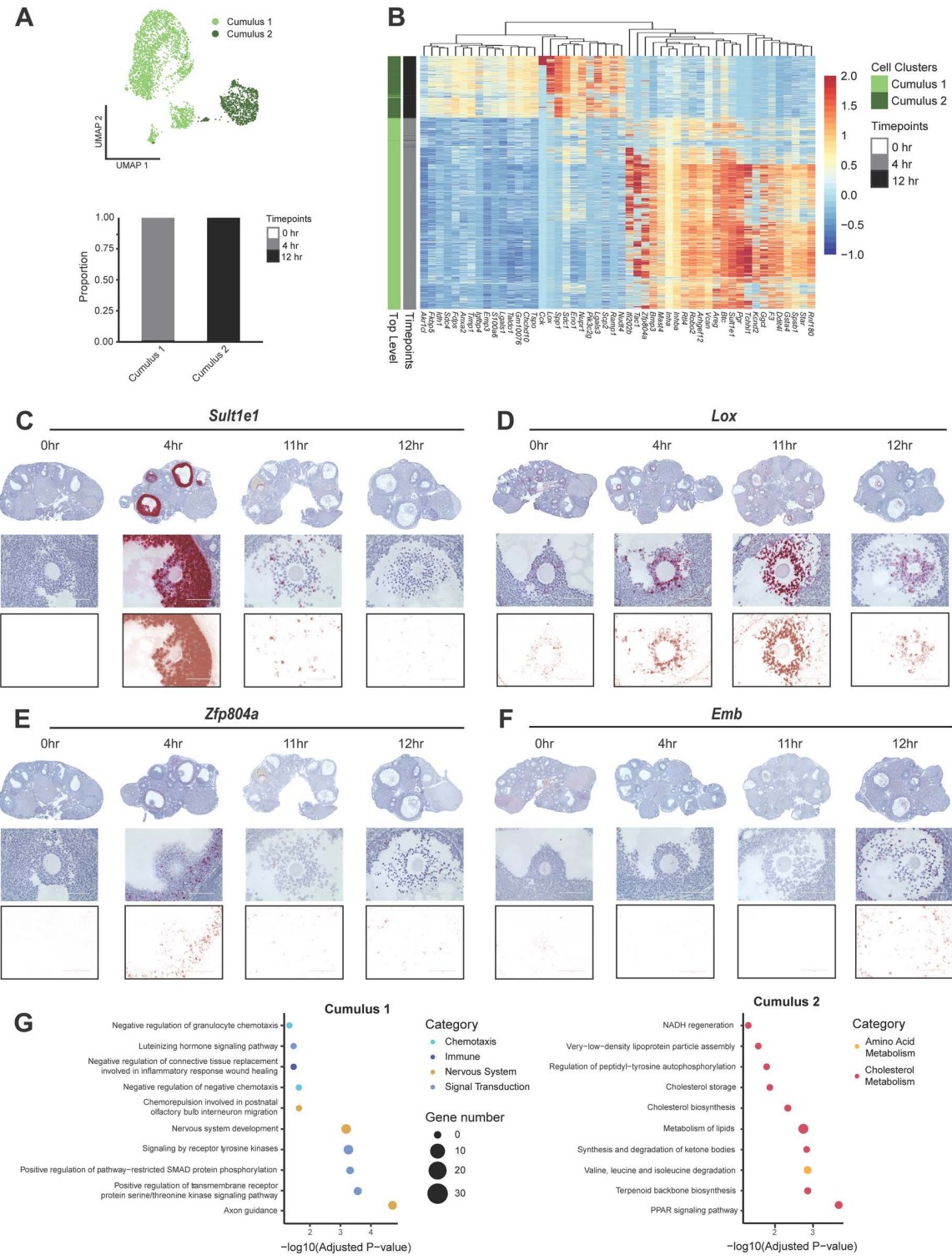

**Fig 5. Cumulus cells exhibit time-dependent changes in gene expression. (A)** UMAP shows the clustering of cumulus cells from scRNA-seq. A stacked bar plot shows the percent of cells in cumulus clusters expressed at each time point. **(B)** Heatmap depicting differential gene expression in Cumulus 1 (early) and Cumulus 2 (late) clusters from scRNA-seq. **(C–F)** RNAscope images show RNA expression of cumulus cell genes of interest, including **(C)** *Sult1e1*, **(D)** *Lox*, **(E)** *Zpf804a*, and **(F)** *Emb*. Top: Ovary scans at 20× magnification. Middle: Brightfield images of COCs at 40×

magnification. Bottom row: Deconvoluted images of COCs at 40× magnification. **(G)** Dot plots show top processes upregulated in early (left) and late (right) cumulus cells. The data underlying this figure is available at the Gene Expression Omnibus (GEO) under accession number GSE294534.

expansion and fluid accumulation within the antral follicle [74–77]. In addition, hyaluronan is secreted by granulosa cells into the follicular fluid, where it contributes to an osmotic gradient that draws fluid from theca cells into the antral space [78,79]. Early cumulus cells also exhibited several pathways related to the nervous system, including "axon guidance" and "nervous system development". Genes that drove the "axon guidance" and "nervous system development" pathways included *Epha5*, *Robo2*, *Epha4*, *Epha7*, *Alcam*, *Sema3c*, and *Efna5* (S1 File). Upregulation of genes important to nervous system function has previously been documented in COCs [73,80], but their role in cumulus cells remains unclear. Late cumulus cells (Cumulus 2) showed enrichment of genes for "positive regulation of prostaglandin biosynthetic process," which is consistent with the role of prostaglandins in stimulating COC expansion [81–84] and promoting dissolution of the COC ECM to facilitate fertilization of the oocyte by sperm [85]. The Cumulus 2 cluster also exhibited several pathways related to cholesterol production and immune function (S1 File).

To determine the pathways that distinguish early cumulus cells from late cumulus cells, we conducted a second GO analysis using the top differentially expressed genes between these two cell populations (S3 File). The transcriptome of early cumulus cells (Cumulus 1) exhibited enrichment of genes related to the nervous system, chemotaxis, immune function, and signal transduction, with the latter representing key pathways driving COC expansion including LH and receptor tyrosine kinase signaling. Late cumulus cells (Cumulus 2) exhibited enrichment in genes related to fatty acid metabolism and cholesterol/hormone synthesis (Fig 5E). Steroid hormone synthesis by cumulus cells has been documented in several species, including mice and humans, and is thought to be important for oocyte maturation and quality [86–94]. Therefore, the upregulation of steroidogenic pathways in late cumulus cells may be supporting several processes that occur near the completion of ovulation, including hormone production in mural cells and fertilization by sperm.

### Analysis of cell types without time-dependent changes also reveal unique expression patterns and putative functions

**Myeloid cells.** For the myeloid cells, we found that this cell cluster primarily consists of macrophages as indicated by expression of known markers (*Cd68*, *Adgre1*, *Fcgr1*, *Cd86*) (S7A Fig). Additionally, we conducted subcluster analysis and identified three subclusters (Subcluster 0, Subcluster 1, Subcluster 2) (S7B and S7C Fig). Although these subclusters did not differ in their temporal profile, they exhibited very different gene expression profiles when comparing their DEGs. Compared to the other subclusters, Myeloid Subcluster 1 is enriched for conventional myeloid markers (*Lyz2*, *C1qa*, *C1qb*, *C1bc*, *Apoe*, *Tyrobp*) as well as pathways involved in antigen presentation, cytokine production, and phagocytosis (S7C–7D Fig and S4 File). During ovulation, macrophages infiltrate the ovary and secrete cytokines, proteases, and vascular growth factors to induce inflammation and support follicle rupture [2,95,96]. In addition, they have been shown to recognize and phagocytose atretic granulosa cells, suggesting a role in clearing up local cell debris in ovulating follicles [96].

The top DEGs for Myeloid Subcluster 0 contained genes that were related to adhesion and pathway analysis showed enrichment for pathways involved in ECM organization, degradation, and signaling (S7C–7D Fig and S4 File). These findings are consistent with the known roles of macrophages and monocytes in producing proteases [97,98], which may contribute to follicle wall and ovarian surface remodeling during ovulation. Macrophages also secrete interleukins such as IL-1A and IL-1B, which induce the expression of proteinases and gelatinases [99,100] that may also be involved in follicle wall or ovarian surface epithelium degradation. Myeloid Subcluster 2 showed high expression of genes involved in hormone signaling within the ovary (*Inhba*, *Inha*, *Fst*, and *Fshr*), and exhibited upregulation of pathways related to ovulation and hormone production (S7C–7D Fig and S4 File). Finally, Myeloid Subcluster 2 was found to have enriched functions for

PLOS Biology

hormone processes, differentiation, and kinase signaling (S7D Fig). These macrophages were the smallest cluster, so it is challenging to draw significant conclusions, but these may play a role in responding to hormone signaling in the ovary.

**Endothelial cells.** For the Endothelial clusters, we conducted DEG and pathway analysis to determine the identity of each cluster (S7 Fig). We found that Endothelial 1 is enriched for genes involved in hematopoiesis (*Pcp4l1*, *Mecom*, *Kitl*) and angiogenesis, including *Flt1* (also known as *Vegfr1*), *Cd34*, and *Nrp1* (S7F Fig). Indeed, pathway analysis of Endothelial 1 shows enrichment in endothelial cell morphogenesis, blood vessel morphogenesis/migration, and angiogenesis (S7G Fig). The importance of ovarian blood vasculature for follicle development and survival, as well as oocyte competence, has been extensively demonstrated in the literature [101–103]. In particular, blood vessels transport key nutrients, signals, and metabolic precursors to the follicle, and deliver waste products away from the follicle [104]. Indeed, we observed co-localization of a marker gene for Endothelial 1 (*Flt1*) cells with the ovarian stroma (*Dcn* – Stroma 1) adjacent to antral follicles at 0 hr post-hCG (S8A Fig). During ovulation, immune cell recruitment to the follicle is mediated by blood vasculature [2]. Follicular fluid originates from blood within the capillaries of the theca externa, and accumulates in the antrum of the follicle in a process fundamental to ovulation [78]. Angiogenesis is also one of the primary processes occurring during corpus luteum development, and these newly formed blood vessels are important for progesterone efflux to support a potential pregnancy [104]. This mechanism is supported by our co-localization data at 12 hr post-hCG, with Endothelial 1 cells remaining distributed within the stroma (with more puncta present) but also within the corpus lutea (S8B Fig). As such, the observed spatial differences during late ovulation aligns with increased ovarian vasculature within the ovarian stroma and corpus lutea.

In contrast to Endothelial 1, Endothelial 2 is enriched for marker genes for lymphatic endothelial cells (*Ccl21a*, *Lyve1*, *Prox1*, *Flt4*) and for pathways involved in lymphatic endothelial cell fate commitment and platelet activation (S7F and S7G Fig). In general, lymphatic vessels are critical for both interstitial fluid homeostasis as well as transporting proteins and immune cells [105]. In the ovary, the lymphatics have known roles in luteolysis [106] and may also transport hormones to and from the bloodstream. Indeed, co-localization of Endothelial 2 (marked by *Gng11*) cells with stroma cells showed spatial and temporal expression patterns similar to Endothelial 1 (S8C and S8D Fig). During ovulation specifically, lymphatic vessels likely contribute to immune cell influx [2,105]. Considering their role in extravascular fluid maintenance, it is also possible that the lymphatics contribute to fluid accumulation within antral follicles [107]. Although these two endothelial clusters do not exhibit temporal differences, their transcriptomic profile indicates that they may facilitate unique roles within the ovary throughout ovulation.

## Cell–cell interaction (CCI) analysis reveals significant interactions that change over the time course of ovulation

To characterize the signaling pathways that are active across ovulation, we conducted cell–cell interaction (CCI) analysis for each cell type at each time point, first using our scRNA-seq dataset (Fig 6A). We observed that the total number of interactions increased over time (S9A Fig). We also found a pattern of distinct cell subclusters that sent and received signals across the ovulation time course (Fig 6A) and that the strength of outgoing and incoming signals was variable for many subclusters (Fig 6B). At 0 hr, we found that the primary senders were Stroma 1, Epithelial, Granulosa 1, and Granulosa 2, where the interaction strength of outgoing signals was high for Stroma 1 and incoming was strong for Endothelial cells. Notably, the cell subcluster that mainly received signals at 0 hr was the Luteal cluster. For the 4 hr time point, the primary senders were Stroma 1, Stroma 2, Cumulus 1, Epithelial, and Theca 1 whereas Mixed luteal, Endothelial 1, and Epithelial were primarily receivers. At the 12 hr time point, the major senders were Cumulus 2, Stroma 1, Stroma 2, and Recent Luteal subclusters (Fig 6A and 6B). These patterns suggest that stromal and cumulus cells remain active communicators, both sending and receiving signals throughout ovulation, in contrast to luteal cells which primarily receive signals from other cell types. Overall, we also observed that stromal cells maintained a similar interaction strength, both incoming and outgoing, over the full-time course of ovulation. We found that across time, outgoing signal strength was dominated by collagen-driven interactions, whereas incoming interaction strengths were mostly mediated by laminin, SPP1, and

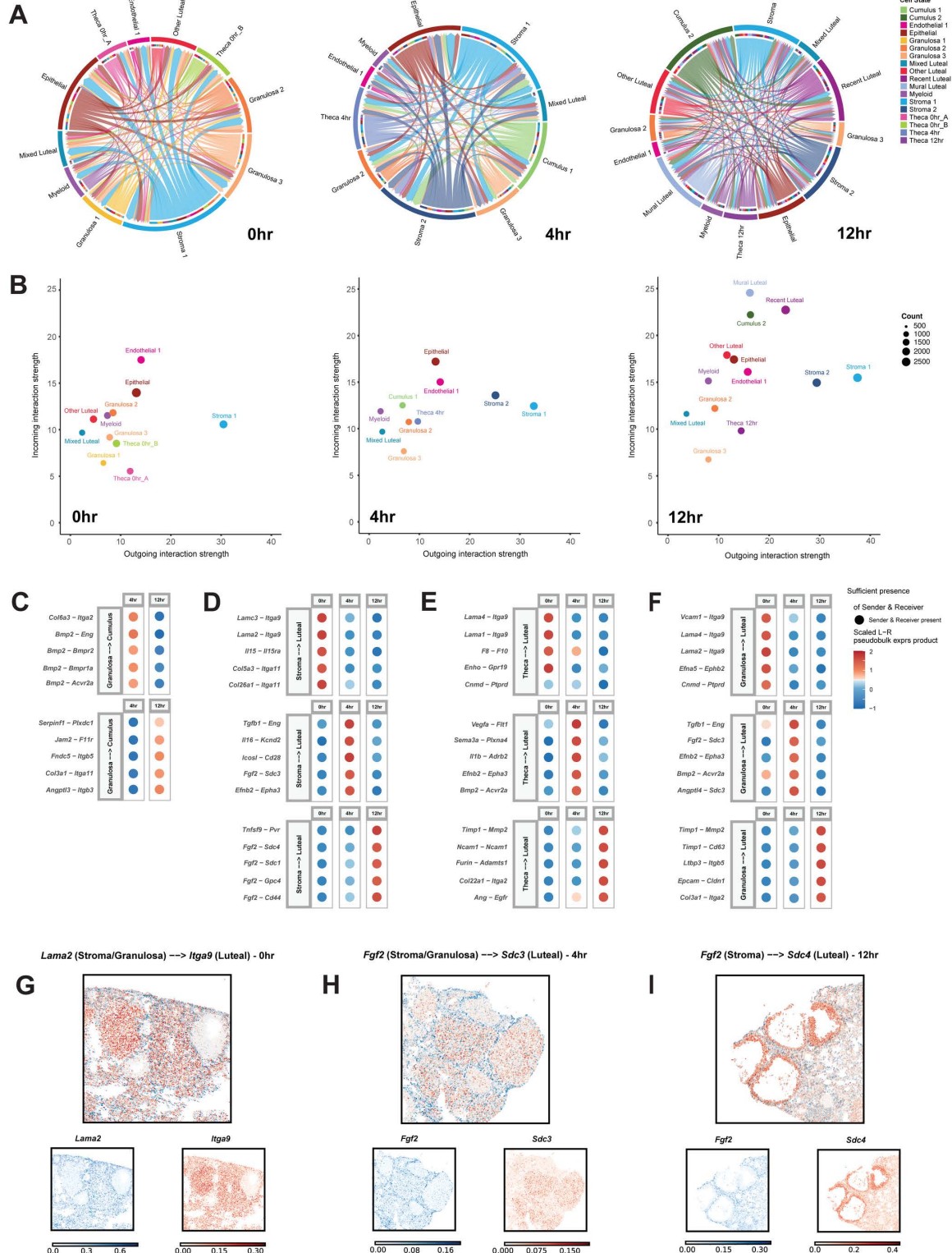

**Fig 6. Cell–cell interactions between cell types change throughout ovulation. (A)** Circle plots show the change of interactions between various cell types at different timepoints. **(B)** Scatterplot shows incoming interaction strength and outgoing interaction strength for all cell types present within the 0 hr (left), 4 hr (middle), and 12 hr (right) timepoints. **(C)** Dot plots show scaled interaction strength between granulosa cells and cumulus cells with

upregulated interactions centered at 0 hr (top) and 4 hr (bottom). **(D)** Dot plots show scaled interaction strength between stroma cells and luteal cells with upregulated interactions centered at 0 hr (top), 4 hr (middle), and 12 hr (bottom). **(E)** Dot plots show scaled interaction strength between theca cells and luteal cells with upregulated interactions centered at 0 hr (top), 4 hr (middle), and 12 hr (bottom). **(F)** Dot plots show scaled interaction strength between granulosa cells and luteal cells with upregulated interactions centered at 0 hr (top), 4 hr (middle), and 12 hr (bottom). (G–I) Expression plots show colocalization of **(G)** *Lama2* and *Itga9* (0 hr), **(H)** *Fgf2* and *Sdc3* (4 hr), and **(I)** *Fgf2* and *Sdc4* (12 hr).

THBS (S9B–9D Fig). Notably, we found the three cell subclusters with the highest incoming signal strength at 12 hr as Cumulus 2, Recent Luteal, and Mural Luteal (Figs 6B and S9). Together, this suggests a highly dynamic ecosystem of cells actively sending and receiving signals across this narrow window of ovulation, and our analysis has identified several of the strongest drivers of these global alterations.

We next wanted to determine if there are specific ligand and receptor pairs that can explain patterns of communication that are unique to our dataset within the ovulatory window. Specifically, we investigated interactions between cumulus, granulosa, theca, stromal, and luteal cells (Fig 6C–6F). These cell types were selected based on the relevance of their interaction to ovulation. For example, communication between granulosa and cumulus cells is critical for processes including granulosa cell differentiation and proliferation, cumulus cell layer expansion, oocyte growth and meiotic progression, and follicle rupture [108–112]. Furthermore, the communication of theca and granulosa cells to luteal cells is essential for the development of new corpora lutea as well as the degradation of existing ones as part of luteolysis. Notably, the stromal clusters observed the highest outgoing strength across all timepoints (Fig 6B) and thus may contribute to the luteinization process, possibly via laminin, collagen, FN1, and several other proteins that show strong outgoing strength (S9B–9D Fig).

At 0 hr post-hCG, we observed an enriched interaction between laminins (ligand), such as *Lama1*, *Lama2*, *Lama4*, and *Lamc4* from granulosa, theca, and stroma cells, and the integrin *Itga9* (receptor) from luteal cells (Fig 6D and 6F). Laminins are known to localize to theca cells to form the basement membrane across all follicle stages [113,114] with additional expression in granulosa and stroma cells [45]. Integrin subunits, such as *Itga9* and *Itb1*, are known to be expressed in the luteal cells, with their interactions with laminins modulating progesterone synthesis during luteinization [115]. Although integrin α9 and its interaction with laminin is largely uncharacterized in the ovary, they are highly localized together (Fig 6G) and suggest that they may play a similar role. At 4 hr post-hCG, interactions between BMP2 and its receptors (BMPR1a, BMPR2, AVCR2A) were highly upregulated between granulosa and cumulus cells, respectively (Fig 6C). BMP2 has been previously shown to be highly expressed in mural granulosa cells in antral follicles [116], whereas loss of BMP receptors has been shown to lead to reduced cumulus cell expansion and spontaneous ovulation [117,118]. At this time point, we also observed communication between theca/granulosa cells and luteal cells via BMP2 and ACVR2A (Fig 6E and 6F). Expression of *Bmp2* and its receptors is known to be suppressed following hCG administration and is re-expressed during luteolysis of corpus lutea, suggesting a role as an inhibitor of luteinization and formation of the early corpus luteum [119]. Additionally, we observed an enriched interaction between *Fgf2* (ligand) and *Sdc3* (receptor) from stroma/granulosa cells and luteal cells, respectively (Fig 6D and 6F), with their co-localization observed spatially (Fig 6H). FGF2 acts as a mitogenic factor in promoting luteal cell proliferation and extracellular matrix remodeling during luteal angioregression [120]. *Sdc3*, which encodes for the proteoglycan syndecan-3, is expressed in the corpus lutea and is suggested to be involved in inflammation [121].

At 12 hr post-hCG, we notably observed enriched signaling of FGF2 from stromal cells to various proteoglycans in luteal cells (Fig 6D), including syndecans and glypican 4. FGF2 is also involved in pro-angiogenic actions in the developing corpus lutea and is highly expressed during the follicular-luteal transition [122,123]. Syndecan-1 and 4 are known to be involved in cytoskeletal reorganization and focal adhesions, respectively, and thus may promote tissue remodeling to generate new corpus lutea [124,125]. Indeed, *Sdc1, Sdc4,* and *Gpc4* have been shown to be enriched in luteinized mural granulosa cells with *Sdc4* being co-localized with *Fgf2* spatially [126] (Fig 6I). In addition, interactions between

*Timp1* (ligand) from theca cells and *Mmp2* (receptor) from luteal cells were highly upregulated at 12 hr post-hCG (Fig 6E). Tissue inhibitor metalloproteinases (TIMPs) and matrix metalloproteinases (MMPs) are well-established to interact and be involved with ovarian remodeling during ovulation, luteinization, and luteolysis [127–130].

## Discussion

Ovulation is a highly coordinated spatiotemporal event that is fundamental to fertilization, reproduction, and endocrine function. To better understand the dynamic changes occurring during ovulation, our study has created an integrated scRNA-seq and iST resource that fully maps this landscape with resolution on the hour timescale. We developed an integrated framework that combines gene expression profiles from single-cell sequencing with spatial information to build a comprehensive map of ovarian cell types, what their functional programs are, and how these cells interact over the full course of ovulation.

While IHC can localize proteins within tissues, it is limited by the availability of validated antibodies, the number of proteins that can be simultaneously detected, and the lack of transcript-level resolution. Additionally, traditional transcriptomic methods such as qRT-PCR, northern blots, and microarrays allow for the targeted detection of selected genes but require investigators to predefine their genes-of-interest, making them unsuitable for unbiased discovery. Although bulk RNA sequencing resolves some of these issues by enabling whole-transcriptome analysis, it still lacks spatial resolution and requires dissociated cells, causing a loss of critical information about the cellular context and microenvironmental influences. These limitations are particularly important in the ovary, where cellular organization and interactions play crucial roles in various biological processes such as follicle development, ovulation, and corpus luteum formation. To address these challenges, recent studies have transitioned towards single-cell and spatial transcriptomic approaches. Single-cell RNA sequencing has been instrumental to investigate ovarian biology, such as a single-cell ovary atlas across the mouse estrous cycle [17], single cells selected across follicle activation [18], and, recently, spatial transcriptomic has been used to characterize transcriptomic profiles in premenopausal adult human samples [27,28]. These studies have been instrumental in delineating cell types in the ovary but have focused on specific regions of interest using GeoMX or provided deconvolved cell states during ovulation which lack single-cell resolution [22,28]. Therefore, our study is unique in its distinctive focus on the temporal dynamics of ovulation across a short time frame, employing both scRNA-seq and iST, with true single-cell resolution, to enable data integration across both datasets. Here, we show successful integration of both data types allowing for inference of spatial expression of over 25,000 genes across the time course of ovulation. With this resource, we interrogated the transcriptional dynamics of ovulation, identified novel gene markers for ovulation-dependent cell types, revealed new biological pathways that may contribute to normal ovarian function, and uncovered novel CCIs that may be important for orchestrating these processes.

First, we identified all expected major cell types of the ovary, including cumulus cells, endothelial cells, epithelial cells, granulosa cells, luteal cells, myeloid cells, stromal cells, theca cells, and oocytes. It is likely that the majority of oocytes were filtered out during single-cell preparation suspension, but immature oocytes may have passed through the filter due to their smaller size, consistent with other sc-RNAseq studies of the ovary [17,19,131]. Mouse oocytes are known to increase in size throughout oogenesis but range from approximately 20 μm (in primordial follicles) to approximately 70 μm at terminal size [132]. Given that the filter used to generate a single cell suspension was 30 μm in diameter, we have likely captured some small oocytes from primordial and primary follicles. Although selecting a filter with a larger diameter would incorporate more oocytes into the suspension, it would also increase the number of doublets, which could artificially alter the observed transcriptomic profiles across cell types. Beyond the limitations of size exclusion, oocytes are not ideal candidates for 10× sequencing methods/droplet-based approaches as they tend not to load well into droplets [133]. In addition, although there are many primordial follicles containing immature oocytes in the mouse ovary, the enzymatic and mechanical digestion techniques used to dissociate ovarian tissue for preparation of single-cell suspensions are harsh on primordial follicles, which may have further reduced the number of oocytes sequenced.

PLOS Biology

A major finding of this study is the identification of time-dependent cell states, between which gene expression profiles were distinct enough to cause these cell states to cluster separately and be enriched for distinct functions and pathways. Our data revealed "early" and "late" subclusters of major ovarian cell types and our results demonstrate that early and late cell clusters can be broadly distinguished by shifts in their functional pathways, where there is a greater diversity of pathway terms in the clusters associated with the later time point in ovulation. Notably, we did not find any time-varying subclusters in myeloid and epithelial cells, which aligns with previous studies evaluating the mouse estrous cycle [19].

Although other transcriptomic investigations in the mouse ovary have identified time-dependent granulosa subclusters across the estrous cycle, such as the prevalence of mural cells in the estrus phase [17], the design of our study afforded increased resolution into granulosa cell types and states. Notably, we observed two time-varying cumulus cell subclusters whereas previous studies found cumulus cells to be clustered together with granulosa cells from preantral follicles or only one cumulus cell cluster across the ovulation time course. These differences may be driven by an increased number of cumulus cells present in our dataset, whether through our hyperstimulated mouse model versus ovulatory follicles naturally found during the estrous cycle, or a higher number of ovaries per time point [17,22]. Notably, the gene signature of the previously identified cumulus cell cluster aligns with our late cumulus cluster (Cumulus 2) [17]. As such, the identification of the early cumulus subcluster (Cumulus 1), which is uniquely enriched for neuronal pathways, may provide additional insight into its function and role in the preovulatory follicle. The detection of time-dependent cumulus cell clusters also has unique relevance for non-hormonal contraceptive discovery as actively expanding cumulus cells are specific to ovulation. Although we did not identify a specific cumulus cell cluster at 0 hr, we find that *Kctd14*, which was previously identified to be enriched in a cluster of preantral granulosa/cumulus cells [17], is also expressed in preantral follicles in our datasets, particularly in our Granulosa 3 cluster (S10 Fig). Thus, 0 hr cumulus cells may be incorporated alongside granulosa cells within the Granulosa 3 cluster.

We further investigated the enriched gene pathways and cell–cell interactions that contributed to the time-dependent changes in ovarian cell states during ovulation. Our findings of upregulated ECM genes and pathways in the Stroma 1 cluster, such as *Dcn and Ogn*, indicate that early stroma cells are primarily involved in ECM formation and maintenance of the tissue architecture as expected from previous literature [134,135]. In contrast, late stromal cells exhibit a variety of pathways including EGF signaling, angiogenesis, and hormone production, supporting a more diverse role for stromal cells in ovulation. Following the LH surge, EGFR signaling is critical for the ovulatory response across ovarian cell types, including stromal vasculature [136,137]. Although hormone biosynthesis is not a classic function of stroma cells, subpopulations of stroma cells capable of producing hormones have been documented in several species [27,138–142]. In theca cells, we identified two populations at the 0 hr time point with one enriched for ovulation genes, such as *Fshr*, *Inhba*, and *Fst*, and another enriched with steroidogenesis, such as *Cyp17a1*. However, later theca subclusters transitioned towards enrichment of pathways involved with signal transduction, immune processes, and metal homeostasis. Similar to stromal cells, late theca subclusters, especially those at the 4 hr time point, were highly enriched for EGFR signaling as expected following the LH surge [136,137]. Late theca cells also upregulated several immune processes, which aligns with its known role in cytokine production and leukocyte recruitment during ovulation [1]. A new insight from our data is the possible role of metal homeostasis in the functions of theca cells. In addition to steroid hormone production, the mitochondria also participate in metal homeostasis by taking in metals and chemically bonding them to proteins (metalation) [143,144]. As steroidogenesis generates reactive oxygen proteins, metalloproteins in theca cells may be involved in reducing reactive oxygen species. Taken together, our results reveal dynamic populations of stromal and theca cells and suggest several understudied functions for these cells during later timepoints in ovulation.

In luteal cells, we observed a shift in the nature of enriched pathways from early to late luteal cell subclusters. Mixed luteal cells, which include those from previous cycles, exhibited upregulation in lipid metabolism which aligns with the highly steroidogenic nature of luteal cells in mature corpus lutea [145]. Complementary to this function include enriched pathways in mitochondrial function, including redox homeostasis and ETC processes. Interestingly, mature corpus lutea

exhibit increased mitochondrial volume and ATPase signal, suggesting that these novel pathways increase respiratory activity to support corpus luteum function. In Recent Luteal cells that are mainly present at 12 hr, we observed upregulation in ECM and smooth muscle pathways. During this time point, luteal cells (composed of luteinized theca and granulosa cells) contribute to ECM remodeling for the formation of new corpus lutea [146]. This observation is further supported by the observed cell–cell interactions between theca or granulosa cells (senders) and luteal cells (receiver) at 12 hr. Specifically, we found enriched interactions that promote tissue remodeling (*Timp1-Mmp2*) and angiogenesis (*Ang-Egfr*). In cumulus cells, we observed strikingly different pathways enriched when comparing the early and late subclusters. Notably, the early cumulus subcluster upregulated pathways relating to nervous system function. Since cumulus cells undergo substantial cellular changes during the LH surge, such as cell migration and retraction of transzonal projections from the oocyte, they may possess neural-like plasticity during expansion [147–150]. However, further evidence is required to elucidate the mechanisms of neuronal pathway genes during ovulation. In contrast, the late cumulus subcluster was enriched with primarily cholesterol metabolism pathways. Recent interactome analyses performed between cumulus and mural granulosa cells as senders and receivers, respectively, revealed a number of interactions related to steroidogenesis [112]. In addition, cumulus cells have been shown to produce progesterone that may partially serve as a sperm chemoattractant following ovulation [151].

This study also revealed three distinct transcriptional identities of myeloid cells present across ovulation. Two of these subclusters were enriched for macrophage functions that are likely related to tissue maintenance, and one possible subcluster that may respond to hormones. It is possible that the hormone responsive macrophages may be due to these myeloid cells phagocytosing other cell types and hence taking on signatures of their local tissue niche. However, an alternative explanation is that myeloid cells may have inherent plasticity which allows them to adopt some of the characteristics of cells they are adjacent to or interact with. This possibility is further supported by our spatial dataset (Figs 1F and S4) which shows that myeloid cells are present across the ovary (stroma, follicles, corpus lutea), as well as our cell–cell interaction analysis (Fig 6A–6B) which indicates that myeloid cells receive most of their incoming signals from stroma cells. It has also been previously shown that the cumulus and the ovarian stroma acquire immune and inflammatory signatures during ovulation [2,73]. Our analyses provide a better understanding of how myeloid cells may transcriptionally change throughout ovulation.

Understanding the spatiotemporal expression of genes in the ovary throughout ovulation has a wide range of implications for infertility and non-hormonal contraceptive development. For example, one condition of particular relevance to this study is luteinized unruptured follicle (LUF) syndrome, in which follicle rupture and COC release do not occur but luteinization and progesterone secretion are maintained [3,152,153]. The cause of LUF syndrome is largely unknown, but LUFs are found in about 11% of women with regular menstrual cycles and up to 43% of women with infertility, with recurrence rates of up to 90% between cycles [4,154]. Because LUF syndrome does not impact hormone secretion, the genes dysregulated in this process may also serve as nascent ovulation-blocking targets that do not affect luteinization. Our study has revealed genes with significant changes in expression within non-luteinizing cells (e.g., cumulus cells) that may be important for orchestrating the morphological and transcriptomic changes that occur within a time span of hours to promote successful ovulation. Thus, this study has the capacity to isolate genes that regulate other key processes in ovulation beyond luteinization, such as COC expansion or follicle rupture, that may drive phenotypes like LUF and be harnessed as non-hormonal contraceptives. Taken together, our integrated dataset presents an array of possible contraceptive or fertility-based targets, representing a large and important resource for the field.

In order to induce ovulation and maximize cell yield, mice were hyperstimulated with PMSG and hCG prior to ovary collection. It is possible that hyperstimulation may impact cell behavior to a different extent than what would occur in a natural ovulation cycle. In addition, because physiologic ovulation in mice occurs overnight, we offset induction of ovulation by 24 hr so that the single cell dissociations could be immediately processed within working hours of the sequencing core. However, we saw no difference in the number of COCs ovulated with the natural or offset ovulation timing (S2B Fig) and

confirmed that ovulation occurred as expected via histology (S1 Fig). Spatial transcriptomics as a novel technology still has various limitations. Cell segmentation is challenging due to the intricate and densely packed cellular environments, which can often lead to false positives and negatives in identifying and distinguishing individual cells. Additionally, the manual selection of probes and the limit to the number of probes restricts our ability to capture a more unbiased spatial profile. On the other hand, single-cell RNA sequencing failed to detect a substantial number of oocytes because the dissociation method was inefficient at capturing larger cells, leading to an underrepresentation of this cell type. Due to the limitations described here, this manuscript is not equipped to be a resource for oocyte biology but instead can be utilized to evaluate the expression profiles of all other major ovarian cell types throughout ovulation.

In summary, this study outlines the dynamic spatiotemporal profile of mouse ovaries across the ovulation time course by combining single-cell resolution with spatial localization. We identified time-varying cell subclusters for major ovarian cell types with enrichment of established and novel markers. Furthermore, we conducted cell–cell interaction analyses between ovarian cell types throughout ovulation, which revealed previously undescribed ligand-receptor interactions. This comprehensive dataset provides the framework to further investigate ovarian cell states during ovulation and may provide implications to better understand anovulatory conditions and drive the discovery of new contraceptive targets for women.

## Materials and methods

### Animals

Female CD1 mice were purchased from Inotiv (West Lafayette, IN, USA) and used when reproductively adult (6–12 weeks). Mice were housed in the Center for Comparative Medicine at Northwestern University within a controlled barrier facility (Chicago, IL, USA) and kept at a constant temperature and humidity in a light cycle of 14 hr light and 10 hr dark. Mice were provided with food and water ad libitum and fed a specific chow that excludes soybean meal (Teklad Global 2916 chow, Envigo, Indianapolis, IN, USA). The Institutional Animal Care and Use Committee at Northwestern University approved all animal experiments (IS00017891), which were also performed in accordance with the National Institute of Health Guide for the Care and Use of Laboratory Animals.

### Optimization of ovulation timing for workflow compatibility

Physiological ovulation in mice typically occurs in the middle of the night which precludes necessary sample processing during normal operating hours of the NU Sequencing core. Therefore, we determined whether we could offset the timing of ovulation in mice by 12 hr without affecting egg yield. Mice were hyperstimulated with pregnant mare serum gonadotropin (PMSG; ProSpec, #HOR-272) 12 hr apart to stimulate follicle growth. Mice were then injected with human chorionic gonadotropin (hCG; Sigma Aldrich, #C1063) 46 hr after relative PMSG injections to induce ovulation, and 14 hr post-hCG injection, ovulated COCs were collected from the oviduct. COCs were denuded of cumulus cells, and the number of MII eggs was compared between the control and offset groups (S2A and S2B Fig). Similar egg numbers were collected across groups demonstrating that ovulation was not impacted by this shift in superovulation timing.

### Generation of single-cell suspensions from mouse ovaries

Ovulation induction was offset by 12 hr as described above for this experiment. As detailed in Fig 1A, mice received an intraperitoneal injection of 5 I.U. PMSG to stimulate follicle growth. A second intraperitoneal injection of 5 I.U. hCG was given 46 hr following PMSG injection to induce ovulation. Ovary dissection occurred 0, 4, or 12 hr post-hCG injection, with two independent operators dissecting ovaries from 3 mice each. One ovary from each mouse was pooled for single-cell isolation, with a total of three ovaries per suspension (labeled ABC or DEF). The contralateral ovary of each mouse was used for MERSCOPE analysis (see below).

The pooled ovaries were cut into quarters using insulin syringe needles and enzymatically digested in 2 mL aMEM-Glutamax supplemented with 1 mg/mL bovine serum albumin (BSA; Sigma–Aldrich, #A3311) and 1×

insulin-transferrin-selenium (Sigma–Aldrich, #1884) containing 40 µg/L liberase DH (Sigma–Aldrich, #05401089001), 0.4 mg/mL collagenase IV (Sigma–Aldrich, #C5138), and 0.2 mg/mL DNAse I (Sigma–Aldrich, #9003-98-9) for 15 min with gently agitation at 37°C and 5% $CO_2$ for 15 min. Ovaries were then mechanically digested via trituration using a 1000P wide bore tip, and the suspension was strained through a pre-wet 30 µm strainer directly into aMEM-Glutamax containing 10% FBS to quench the enzymes. Any remaining pieces of ovary were returned to the incubator in 2 mL fresh digestion media for another 15 min, followed by repeat mechanical digestion and straining. Once the enzymatic and mechanical digestions were complete, the cell suspension was centrifuged at 300 × g for 10 min at 4°C. The supernatant was removed, and the remaining cell pellet was resuspended in 100 µL Red Blood Cell Lysis solution (Miltenyi Biotec, #130-107-677) and incubated for 10 min at 4°C. After red blood cell removal, the suspension was centrifuged again at 300 × g for 10 min at 4°C. Following supernatant removal, the resulting pellet was resuspended in 100 µL of 0.025% BSA in phosphate-buffered saline without calcium or magnesium, and transferred to lo-bind Eppendorf tubes. The suspension was placed on ice and transferred immediately to the Northwestern University Sequencing Core (Chicago, IL, USA).

**Single-cell library preparation and sequencing**

Cell number and viability were analyzed using Nexcelom Cellometer Auto2000 with AOPI fluorescent staining method. Sixteen thousand cells were loaded into the Chromium iX Controller (10X Genomics, PN-1000328) on a Chromium Next GEM Chip G (10X Genomics, PN-1000120), and processed to generate single-cell gel beads in the emulsion (GEM) according to the manufacturer's protocol. The cDNA and library were generated using the Chromium Next GEM Single Cell 3′ Reagent Kits v3.1 (10X Genomics, PN-1000286) and Dual Index Kit TT Set A (10X Genomics, PN-1000215) according to the manufacturer's manual. Quality control for the constructed library was performed by Agilent Bioanalyzer High Sensitivity DNA kit (Agilent Technologies, 5067-4626) and Qubit DNA HS assay kit for qualitative and quantitative analysis, respectively. The multiplexed libraries were pooled and sequenced on Illumina Novaseq6000 sequencer with 100 cycle kits using the following read length: 28 bp Read1 for cell barcode and UMI, and 90 bp Read2 for transcript. The single-cell library preparation and sequencing were performed at the Northwestern University NUseq core facility with the support of an NIH Grant (1S10OD025120).

**Single-cell RNA-seq (scRNAseq) analysis**

Raw sequencing data was demultiplexed using Cell Ranger from 10× Genomics, converting the raw data into FASTQ format. Cell Ranger was also used for the alignment of the FASTQ files to the reference genome and counting the number of reads from each cell that aligned to each gene. R version 4.2.2 and Seurat version 4 were used for all downstream analyses unless specified otherwise [24]. For initial quality filtering, cells with greater than 20% mitochondrial gene expression (percent.mt) and less than 3000 expression counts (n_FeatureRNA) were removed. The data did not show any significant batch effects. Standard Seurat pipelines were used to scale, find variables, and normalize the dataset. The identified list of variable genes was used to perform the principal component analysis. Cell clusters were identified with the Find-Neighbors function with dims = 1:20 and the Find Clusters function with resolution = 0.5. Subclusters were found with iterative clustering with different resolutions of 0.5, 0.1, and 0.3. DoubletFinder_v3 was used to remove doublets with an approximate 5% expected ratio [155]. MAST was used to perform all differential expression analyses [156]. Cell–cell interaction analysis was performed with CellChatDB and Multinichenet packages [157,158].

**Gene ontology analysis**

The online knowledgebase published by the Gene Ontology Consortium (GO Enrichment Analysis Tool), and several gene set enrichment analysis tools available via Enrichr (Reactome 2022, KEGG 2021 Human, GO Biological Process 2023) were utilized to perform GO analyses on each cell cluster [159–162]. Gene IDs for the top 150 genes upregulated in each cluster were inputted into each tool. For the GO Enrichment Analysis Tool, "biological process" and "mus musculus"

were selected as search filters. GO results from Enrichr and the GO Enrichment Analysis Tool were sorted by ascending adjusted $p$-value and descending fold enrichment, respectively, and only pathways with adjusted $p$-values < 0.05 were included in the analysis. Pathways highlighted in this study were selected from the top 10 pathways from Enrichr or the GO Enrichment Analysis Tool. Selected pathways were annotated by category based on the biological function consistent with the pathway and cell type.

### Preparation of ovaries and microarray assembly for multiplexed error-robust fluorescence in situ hybridization (MERSCOPE) analysis

Ovaries intended for MERSCOPE analysis were collected in RNAse-sterile conditions as intact whole ovaries and stored in cryo-safe tubes and flash frozen in liquid nitrogen before shipping to the Broad Institute for TMA construction and MERSCOPE processing (S2C Fig). In RNAse-sterile conditions, samples were embedded and frozen in a pre-formed scaffold of Optimal Cutting Temperature media, oriented so that the ovarian hilum was at the base of the microarray and stored at −80°C until sectioning. The ovaries were assembled in this tissue-microarray (TMA) as three whole ovaries per each time point (0 hr, 4 hr, and 12 hr) for a total of 9 ovaries. 10 µm-thick sections of the TMA were obtained using a cryostat at −20°C, mounted on fluorescent microsphere-coated, functionalized coverslips, fixed in 4% PFA in 1X PBS, and permeabilized in 70% ethanol overnight (S2D Fig).

Following permeabilization in 70% ethanol, the TMA section was stained with Vizgen's Cell Boundary Stain Kit (PN 10400009). The section was washed briefly with 1X PBS before being incubated for 1 hr and room temperature in 100 µL of Cell Boundary Blocking Buffer Premix (PN 20300012) and 5 µL of murine RNase inhibitor, with a 2 × 2 cm square of parafilm over the sample to spread the mixture and prevent drying. The section was then incubated for another hour at room temperature in a mixture of 100 µL Cell Boundary Blocking Buffer, 5 µL RNase inhibitor, and 1 µL of Cell Boundary Primary Stain Mix (PN 20300010) with parafilm as described above. The section was washed three times with 5 ml 1X PBS on a rocker before a final 1-hr incubation at room temperature in 100 µL Cell Boundary Blocking Buffer Premix, 5 µL RNase inhibitor, and 3 µL Cell Boundary Secondary Stain Mix (PN 20300011) with parafilm as described above. The section was washed three times with 5 ml 1X PBS on a rocker at room temperature, then fixed again in 4 ml of 4% PFA in 1X PBS, followed by two 5 ml 1X PBS, all at room temperature. To hybridize the MERSCOPE probes to the section, the sample was first briefly washed in 2X saline-sodium citrate (SSC) at room temperature, and incubated in 30% formamide in 2X SSC at 37°C for 30 min. An amount of 50 µL of the probe mixture and 1 µL of RNase inhibitor were added on top of the sample and covered with parafilm as described above, and the section was incubated for 48 hr at 37°C. After two 30 min incubations at 37°C in 30% formamide in 2X SSC, the sample embedded in a polyacrylamide gel solution (3.9 ml nuclease-free water, 0.5 ml 40% acrylamide/bis solution 19:1, 0.3 ml 5 M NaCl, 0.3 ml Tris pH 8, 25 µL 10% w/v ammonium persulfate in nuclease-free water, 2.5 µl of N,N,N′,N′-tetramethylethylenediamine). To embed the section, excess formamide was first removed with a 2-min incubation of 2X SSC while the gel mixture was prepared. Once the gel was prepared, the sample was incubated in 5 ml of the gel mixture for one minute. Then the sample was transferred to a clean petri dish, the excess gel mixture was wicked away with a Kimwipe, and 50 µL of reserved gel mixture was added on top of the section. A 20 mm glass coverslip that was cleaned and treated with 50 µL of GelSlick was inverted on top of the sample, spreading out the gel evenly. The excess gel mixture was wicked away with a Kimwipe, and the sample was left at room temperature for 2 hr while the gel was set. To clear excess proteins from the sample, the 20 mm coverslip was removed after the gel had completely set, and the sample was incubated in 5 ml of clearing mixture (3.4 ml nuclease-free water, 1 ml 10% sodium dodecyl sulfate (SDS), 0.5 ml 20X SSC, and 0.1 ml 25% Triton-X) for 3 days at 37°C.

### Selection of MERSCOPE genes

A MERSCOPE panel of 198 genes consisting of marker genes, genes known to be involved in ovulation, and additional genes-of-interest were constructed based on published literature or preliminary data from our lab and others (S2 File).

Marker genes were chosen to facilitate the identification of cell types including granulosa, luteal, germ, mesenchymal, endothelial, epithelial, and immune cells [17,18].

## Construction of MERSCOPE probes

A fluorescently tagged oligo probe library for 198 combinatorial genes and 3 sequential genes was designed, with each probe encoding the barcodes assigned to specific target RNA transcripts in the library. RNA targets were selected based on increasing the success of probe binding and ensuring gene expression falls within optically appropriate parameters for MERSCOPE imaging. Manufacturing services occurred at Vizgen.

## MERSCOPE imaging

The cleared sample was briefly washed three times with 5 ml of 2X SSC, stained with 3 ml of Vizgen DAPI and PolyT Staining Reagent (PN 20300021) for 10 min on a rocker, incubated in 30% formamide in 2X SSC for 15 min, and then transferred to 2X SSC while the MERSCOPE instrument was prepared. The MERSCOPE flow chamber was cleaned with RNaseZap RNase Decontamination Solution and 70% ethanol. A Vizgen MERSCOPE 300 Gene Imaging Cartridge (PN 20300017) was thawed and activated by adding 250 µL of Vizgen Imaging Buffer Activator (PN 20300022) and 100 µL of RNase inhibitor. 15 ml of mineral oil was added on top of the imaging solution in the cartridge to prevent oxidation. The MERSCOPE was initialized, and primed, the section was loaded into the flow chamber, and the flow chamber was attached to the MERSCOPE fluidics system and wetted, checking for bubbles before proceeding. A 10X overview was first acquired of the entire imageable area, and regions of interest were selected before moving to the 60X, which was cleaned and oiled before acquiring the MERSCOPE images. Once imaging was complete, a cell boundary stain was selected for cell segmentation and image analysis was performed on the MERSCOPE using Vizgen's MERlin pipeline to acquire transcript count and cell segmentation data.

## MERSCOPE analysis

Cell segmentation was performed on the MERSCOPE using Vizgen's provided pipeline, which utilizes CellPose and MERlin to acquire cell and transcript data [159,163]. Python version 3.7.12 was used to perform all analysis unless specified otherwise. Cells with less than 10 transcript counts were removed. Scanpy was used to find the variables, normalize, scale, perform PCA, find neighborhoods, and cluster cells with Leiden [164]. To identify the cell type for each cluster, we counted the number of transcripts for each gene in every cluster and looked for known markers in the top 10. We also performed differential expression analysis using Seurat v3 in R to find markers for each Leiden cluster [24]. Squidpy along with anndata and scanpy was used to visualize the regions spatially [164,165]. Cell–cell interaction analysis was performed with CellPhoneDB [166].

## MERSCOPE-10X integration analysis

Integration of MERSCOPE and 10X single-cell datasets was performed using the package Tangram [25]. Out of the 198 genes in the MERSCOPE gene probes, 18 test genes were randomly selected to be left out of the training process, including *Hmgcs2*, *Col1a2*, *Ly6e*, *Krt18*, *Nr2f2*, *Oca2*, *Sult1e1*, *Ccr7*, *Ptprc*, *Cd14*, *Wnt5a*, *Kitl*, *Pbk*, *Rasd1*, *Folr1*, *Rnd3*, *Mro*, and *Cldn5*, to later assess the performance of the integration. The learning model uses the leave-one-out validation strategy, where the remaining 180 genes were partitioned into 179 training genes and a single validation gene. The algorithm repeated the training 180 times, each time leaving out a different one, to obtain a prediction for each gene. The overall performance of the analysis is evaluated in three ways: (1) training and testing scores were obtained to quantify the deep learning model performance, (2) comparison of gene expression from those randomly selected from the test set between the MERSCOPE dataset and their predicted expression based on the integration algorithm, and (3) validation

of predicted gene expression patterns from integrated algorithm with RNAscope images using genes not originally in the MERSCOPE probes.

## Histological processing and staining

Immediately after collection, mouse ovaries were placed in tubes containing 1 mL Modified Davidson's solution (Electron Microscopy Services, Hatfield, PA) and rocked for 2–4 hr at room temperature. Ovaries were then stored overnight at 4°C with gentle rocking. The next morning, ovaries were washed in 70% ethanol three times with 10 min per wash. Using standard processing protocols, an automated tissue processor (Leica Biosystems, Buffalo Grove, IL) was used to process, dehydrate, and embed the ovaries in paraffin wax. Ovaries were then serially sectioned at 5-μm-thick intervals until the approximately half of the ovary was sectioned, and sections were placed on glass slides. The slide containing the approximate midsection of the ovary was stained with hematoxylin and eosin using a standard hematoxylin–eosin staining protocol. Stained sections were cleared using three 5-min-incubations with Citrisolv (Decon Laboratories, King of Prussia, PA) and then mounted with Cytoseal XYL (ThermoFisher Scientific).

## RNA in situ hybridization

The expression of mouse *Sult1e1* (ACD, #900181), *Lox* (ACD, #425311), *Emb* (ACD, #462011), *Zfp804a* (ACD, #1161171-C1), *Trpv4* (ACD, #406071), *Sik3* (ACD, #526431), and *Slc6a6* (ACD, #544751) in whole ovary sections was detected using the RNAscope 2.5 HD Red Assay (Advanced Cell Diagnostics (ACD)). Samples were incubated with positive (*Ppib*; ACD, #313911) and negative (*Dapb*; ACD, #310043) control probes or target probes for 2 h at 40°C and counterstained with hematoxylin and ammonia water. The EVOS FL Auto imaging system was used to scan whole ovary sections at 20× resolution and image COCs at 40× magnification. Images were then color deconvoluted using Fiji [167] to improve signal visibility. The full, stepwise protocol for this assay can be found on the ACD website (https://acdbio.com/sites/default/files/322360-USM%20RNAscope%202.5%20HD%20RED%20Pt2_11052015.pdf).

## Supporting information

**S1 Fig. Hematoxylin–Eosin-stained sections of ovaries collected during in vivo ovulation time course, related to Fig 1. Examples of ovaries collected (A) 0 hr, (B) 4 hr, or (C) 12 hr post hCG injection (left) and insets labeling key cell types (right).** O = oocyte, CC = cumulus cells, GC = granulosa cells, TC = theca cells, SC = stroma cells, LC = luteal cells, Epi = epithelial cells. Scale bars = 200 μm.
(TIFF)

**S2 Fig. Schematic of optimization protocols for ovary collection (A–B) and MERFISH (C–D), related to Fig 1 and "Materials and methods". (A)** Table depicting timing of hormone (PMSG and hCG) injections and ovary collection in control and offset timing groups. **(B)** Graph showing the average number of COCs collected per mouse in control and offset timing groups. **(C)** Top row: Pregnant mare serum gonadotropin (PMSG) and human chorionic gonadotropin (hCG)-stimulated, flash-frozen mouse ovaries were collected as intact whole ovaries or contralaterally-halved ovaries. Samples were embedded into a pre-formed OCT tissue microarray scaffold (TMA) with the ovarian hilums (indicated by red asterisk, *) pointing downward towards the tissue microarray base, enabling uniform tissue section collection onto fluorescent microsphere-coated functionalized coverslips. Bottom row: corresponding images of bisected ovaries, embedding into the TMA scaffold, and 10× DAPI imaging of the resulting tissue sections. **(D)** MERFISH protocol of mounted samples undergoing a series of staining and incubations (fixation, permeabilization, hybridization, polyacrylamide gel embedding, tissue clearing with detergents).
(TIFF)

**S3 Fig. Quality checks, related to** Fig 1. **(A–B) Quality metrics for spatial transcriptomics dataset: bar plots on number of transcripts per cell and cell volume post-filtering. (C)** UMAP of the single-cell dataset, including unknown clusters. **(D)** Quality metrics for single-cell dataset: Violin plots on genes per cell and percent mitochondrial in the single-cell dataset post-filtering. **(E)** Clustered dot plot compares cell types found in both datasets using established markers. The data underlying this figure is available at the Gene Expression Omnibus (GEO) under accession number GSE294534.
(TIFF)

**S4 Fig. Clustering and cell identification of spatial transcriptomic dataset, related to** Fig 1. **(A) Expression plots of known markers for cell types identified in all ovaries at three timepoints show corresponding localization. (B)** Ovaries serve as biological replicates that are not included in the main figures, colored by cell types identified. **(C)** Heatmap showing three top marker genes used to determine the identity of each cell cluster for each ovary in iST. The data underlying this figure is available at the Gene Expression Omnibus (GEO) under accession number GSE294534.
(TIFF)

**S5 Fig. Integration analysis of single-cell and spatial transcriptomics, related to** Fig 1 **and "Materials and methods". (A)** Bar plots of training scores for training genes (left) and scatter plot of test scores versus sparsity for test genes (right), for the three integrations at 0 hr (top), 4 hr (middle), and 12 hr (bottom). **(B)** Expression plots of *Col1a2* for predicted and observed results show similar patterns. **(C)** RNAscope images (left) and predicted results from integration analysis (right) show similar expression patterns. Scale bars = 200 μm. The data underlying this figure is available at the Gene Expression Omnibus (GEO) under accession number GSE294534.
(TIFF)

**S6 Fig. Co-localization of theca or granulosa and luteal cells during early and late timepoints of ovulation.** Expression plot shows colocalization of **(A)** *Cyp17a1* (Blue: Theca 1) and *Gm2a* (Red: Luteal 1) at 0 hr, **(B)** *Col4a1* (Blue: Theca 2) and *Fndc3b* (Red: Luteal 2) at 12 hr, **(C)** *Inhbb* (Blue: Granulosa 1) and *Gm2a* (Red: Luteal 1) at 0 hr, and **(D)** *Amh* (Blue: Granulosa 3) and *Fndc3b* (Red: Luteal 2) at 12 hr. Antral follicles are highlighted by dashed circles. The data underlying this figure is available at the Gene Expression Omnibus (GEO) under accession number GSE294534.
(TIFF)

**S7 Fig. Myeloid and endothelial cell types throughout ovulation. (A)** Feature plots show expression of macrophage markers (*Cd68*, *Adgre1*, *Fcgr1*, *Cd86*) within the myeloid cluster from scRNA-seq. **(B)** UMAP shows the clustering of myeloid cells at the subcluster level from scRNA-seq. **(C)** Heatmap depicts differential gene expression in myeloid subclusters (Subcluster 0, 1, and 2) from scRNA-seq. **(D)** Dot plots show top processes upregulated in Myeloid Subcluster 0 (left), Subcluster 1 (middle), and Subcluster 2 (right). **(E)** UMAP shows the clustering of endothelial cells at the top-level from scRNA-seq. **(F)** Heatmap depicts differential gene expression in Endothelial 1 and Endothelial 2 clusters from scRNA-seq. **(G)** Dot plots show top processes upregulated in Endothelial 1 (left) and Endothelial 2 (right) clusters. The data underlying this figure is available at the Gene Expression Omnibus (GEO) under accession number GSE294534.
(TIFF)

**S8 Fig. Co-localization of endothelial and stromal cells during early and late timepoints of ovulation.** Expression plot shows colocalization of **(A)** *Flt1* (Blue: Endothelial 1) and *Dcn* (Red: Stroma 1) at 0 hr, **(B)** *Flt1* (Blue: Endothelial 1) and *Pdgfra* (Red: Stroma 2) at 12 hr, **(C)** *Gng11* (Blue: Endothelial 2) and *Dcn* (Red: Stroma 1) at 0 hr, and **(D)** *Gng11* (Blue: Endothelial 2) and *Pdgfra* (Red: Stroma 2) at 12 hr. Antral follicles are highlighted by dashed circles. The data underlying this figure is available at the Gene Expression Omnibus (GEO) under accession number GSE294534.
(TIFF)

**S9 Fig. Outgoing and incoming cell–cell interactions, related to** [Fig 6]. **(A) Bar plot of total number of interactions per ovulation time point. (B)** Heatmap of incoming and outgoing signal patterns at the 0 hr time point. **(C)** Heatmap of incoming and outgoing signal patterns at the 4 hr time point. **(D)** Heatmap of incoming and outgoing signal patterns at the 12 hr time point. **(E–G)** Expression plots show colocalization of the whole ovary for **(E)** *Lama2* and *Itga9* (0 hr), **(F)** *Fgf2* and *Sdc3* (4 hr), and **(G)** *Fgf2* and *Sdc4* (12 hr). The data underlying this figure is available at the Gene Expression Omnibus (GEO) under accession number GSE294534.
(TIFF)

**S10 Fig. Expression of *Kctd14* within integrated single-cell and spatial transcriptomic dataset. (A)** Inferred expression of *Kctd14* from the integration of scRNA-seq and iST data shows localization to preantral follicles. **(B)** Violin plot shows *Kctd14* expression in granulosa cell clusters, including Granulosa 3. The data underlying this figure is available at the Gene Expression Omnibus (GEO) under accession number GSE294534.
(TIFF)

**S1 File. Pathway analysis for top-level clusters, related to Fig 1.**
(XLSX)

**S2 File. List of MERFISH probes.**
(XLSX)

**S3 File. Differentially expressed genes between time-dependent cell states, related to Figs 2–5.**
(XLSX)

**S4 File. Marker genes for subclusters, related to Figs 2–5 and S7.**
(XLSX)

**S5 File. Pathway analysis results for subclusters, related to Figs 2–5 and S7.**
(XLSX)

## Acknowledgments

We would like to thank Hoi Chang Lee and Yiru Zhu for their assistance in collecting the ovary samples utilized in this study, and Elizabeth Tsui for providing a protocol for dissociation of single cells from the ovary that we adapted for this study. We also thank the Northwestern University Sequencing Core Facility for conducting the single-cell sequencing of our samples.

## Author contributions

**Conceptualization:** Jeffrey Pea, Francesca E Duncan, Brittany A. Goods.

**Data curation:** Ruixu Huang.

**Formal analysis:** Ruixu Huang, Caroline E Kratka, Jeffrey Pea.

**Funding acquisition:** Alex K Shalek, Brian Cleary, Samouil L Farhi, Francesca E Duncan, Brittany A. Goods.

**Investigation:** Ruixu Huang, Caroline E Kratka, Cai McCann, Jack Nelson, Daniela D Russo, Emily J Zaniker-Gomez, Achla H Gandhi.

**Methodology:** Ruixu Huang, Caroline E Kratka, Cai McCann, Luhan T Zhou, Alex K Shalek, Samouil L Farhi, Francesca E Duncan, Brittany A. Goods.

**Project administration:** Jeffrey Pea, Alex K Shalek, Samouil L Farhi, Francesca E Duncan, Brittany A. Goods.

**Resources:** Alex K Shalek, Samouil L Farhi.

**Software:** Ruixu Huang, John P Bryan, Brian Cleary.

**Supervision:** Alex K Shalek, Francesca E Duncan, Brittany A. Goods.

**Validation:** Ruixu Huang, Caroline E Kratka.

**Visualization:** Ruixu Huang, Caroline E Kratka, Jeffrey Pea, John P Bryan, Francesca E Duncan, Brittany A. Goods.

**Writing – original draft:** Ruixu Huang, Caroline E Kratka, Jeffrey Pea, Francesca E Duncan, Brittany A. Goods.

**Writing – review & editing:** Ruixu Huang, Caroline E Kratka, Jeffrey Pea, Cai McCann, Jack Nelson, John P Bryan, Luhan T Zhou, Daniela D Russo, Emily J Zaniker-Gomez, Achla H Gandhi, Alex K Shalek, Brian Cleary, Samouil L Farhi, Francesca E Duncan, Brittany A. Goods.

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
