## [Editor Report · Decision Letter 0]

12 Sep 2024

Dear Dr Goods, 

Thank you for submitting your manuscript entitled "Single-cell and spatiotemporal profile of ovulation in the mouse ovary" for consideration as a Methods and Resources by PLOS Biology. I'm sorry for the delay in contacting you with a decision, due to a busy time for the journal.

Your manuscript has now been evaluated by the PLOS Biology editorial staff as well as by an academic editor with relevant expertise and I am writing to let you know that we would like to send your submission out for external peer review.

Once your full submission is complete, your paper will undergo a series of checks in preparation for peer review. After your manuscript has passed the checks it will be sent out for review. To provide the metadata for your submission, please Login to Editorial Manager (https://www.editorialmanager.com/pbiology) within two working days, i.e. by Sep 14 2024 11:59PM.

Kind regards,

Suzanne

Suzanne De Bruijn, PhD, 

Associate Editor

PLOS Biology

sbruijn@plos.org

---

## [Decision Letter · Decision Letter 1]

9 Dec 2024

Dear Dr Goods,

Thank you for your patience while your manuscript "Single-cell and spatiotemporal profile of ovulation in the mouse ovary" was peer-reviewed at PLOS Biology. It has now been evaluated by the PLOS Biology editors, an Academic Editor with relevant expertise, and by several independent reviewers. In light of the reviews, which you will find at the end of this email, we would like to invite you to revise the work to thoroughly address the reviewers' reports.

As you will see below, the reviewers comment that the study offers a useful and comprehensive dataset. However they have also provided a number of suggestions to strengthen the study further and to make some of the findings clearer and we think these points should be carefully addressed before publication. As a last note, we would not require that you substantially expand the study with new analyses, as suggested by Reviewer 2. Reviewer 2 has suggested that you develop the study further by analyzing the data more deeply and/or by showing how the current study and approach goes beyond previous characterizations. While we think this certainly would be interesting, after discussion with the Academic Editor and the other reviewers, we think the study and it's results already fits within the scope of our 'Methods and Resources' article type, and we would not require that it be substantially expanded. 

However, given the extent of the other revisions needed, we cannot make a decision about publication until we have seen the revised manuscript and your response to the reviewers' comments. Your revised manuscript may be sent for further evaluation by all or a subset of the reviewers.

**IMPORTANT - SUBMITTING YOUR REVISION**

*Re-submission Checklist*

*Published Peer Review*

*PLOS Data Policy*

*Blot and Gel Data Policy*

Sincerely,

Luke

Lucas Smith, Ph.D.

Senior Editor

PLOS Biology

lsmith@plos.org

REVIEWS:

Reviewer #1, David Pépin (note, reviewer 1 has signed this review): Major comment:

Throughout the manuscript it would be advantageous for each major cell type (stromal, theca, luteal etc.) to include a subset of DEG markers defining each cluster (say top5-10 by p-value or LogFC in a heatmap, violinplot, dotplot or equivalent) in the corresponding figure or as tables. It would make the manuscript a more straightforward resource when quickly comparing gene lists. It would also make it immediately more intuitive what the genes chosen to be validated correspond to.

In Fig3- the nomenclature of theca1 and theca 2 as early and late, and subcluster naming of 0h-theca1 and 0h-theca2 could lead to confusion, perhaps rename the latter. Do they correspond to immature and steroidogenic theca? There is no exploration of the DEG between 0h-theca1 and 0h-theca2. The clustering in B makes it difficult to identify the salient markers for each cluster type.

Line 287: Do you have evidence the cluster includes both granulosa and theca cells differentiating into luteal cells rather than just granulosa cells?

Line 349: Cumulus 1 and cumulus 2 do not include any cells from the 0h condition. It is likely that the "0h" cluster was too different to be picked up as a distinct cluster within the granulosa cell supercluster. Cumulus cells are transcriptionally almost identical to preantral granulosa cells, whereas antral mural gc will be closer to the luteinizing mural. Your MERFISH likely includes examples of this, if looking at markers like Kctd14.

Fig 5 - This figure doesn't have the temporal expression violin plots you have in the previous figures, presumably because the cumulus 0h wasn't identified. Still, the message would be easier to understand with the corresponding expression data for the targets chosen for validation across those two timepoints. 

There is a conspicuous absence of discussion of the myeloid and endothelial clusters despite their relatively large representation in the dataset. Given their potential role in ovulation, it would be of interest to have at least some superficial analysis of DEG across timepoints. Even fewer than expected differences would be interesting.

Could you include a list of probes used for MERFISH.

Minor comment:

In fig1- I greatly appreciate the consistent color scheme for the cell clusters, but panel 1C could still use headings at the top. 

Reviewer #2: The authors comprehensively analyzed how gene expression patterns change in various ovarian cells during ovulation by single-cell RNAseq and MERSCOR techniques. Although there have been reports of single cell RNAseq in all ovarian cells during the estrus cycle, there have been no reports of detailed time-dependent analysis of the ovulation process, so this database is considered to be useful for clarifying ovulation phenomena from multiple perspectives. In addition, the value of this database is further enhanced by the fact that the MERSCOR technique simultaneously shows where the cell groups of each cluster analyzed by single cell RNAseq are localized. 

On the other hand, the results of this analysis are almost identical to those of granulosa cells and cumulus cells that have been performed by microarray analysis and RNAseq for more than 20 years. This indicates the appropriateness of the experimental method of this study by combining single cell RNAseq and MERSCOR technology. However, it can be said that new academic knowledge is scarce. 

However, unlike the exhaustive detection methods of the past, this research method should be able to dig deeper into the differences between cells within each cluster at each sample time. Based on the diversity and localization analysis within each cluster, it is expected that more detailed analysis will be possible, such as the difference in expression patterns between the cumulus cells in the vicinity of the oocytes and the follicular cavity, and the expression patterns in granulosa cells differ between the follicular membrane side and the follicle cavity side. Unfortunately, this paper only makes a rough classification, such as dividing cumulus cells into different clusters in the early and late ovulation phases.

If this paper is to be a database or methodological paper, the differences from past detection methods should be shown by dividing them into advantages and disadvantages. If researchers combine existing transcriptome analysis with IHC, they may be able to obtain similar experimental results. How do we answer this question?

In addition, the experimental method and its value as a database have not been improved by the GO analysis and cell-cell interaction analysis conducted in this paper. If a function prediction is to be made, it should be clarified that the prediction is correct and what its significance is. In addition, the reviewer thinks that such results can be obtained by separating each cell and performing RNAseq, and in fact, several such papers have been published.

The reviewer thinks that this paper should focus on demonstrating the correctness of the experimental method and its value as a database. Otherwise, the reviewer would have to point out that the academic value and novelty of the analysis results would be limited.

Reviewer #3: Comments for Huang et al. "Single-cell and spatiotemporal profile of ovulation in the mouse ovary " 

Comments to Authors

In this MS, the Authors through scRNA-seq and iST analyses, we have captured major ovarian cell types and their spatial distributions across 3 time points around stimulated ovulation in CD1 mice. Overall, it a comprehensive study describing the events of ovulation in the mouse and serves as a robust resource for discovering cell states that are unique to ovulation. Furthermore, the focus on the temporal dynamics of ovulation across a short time frame, employing both scRNA-seq and iST is elegant and innovative and the experimental approach of utilizing one ovary from each mouse for scRNA-seq and the contralateral ovary for iST at each of the specific timepoints is elegant and powerful. This study is acceptable after some corrections and clarifications are made. Please find my comments below:

1. Fig 1F - the color chosen for theca cells is difficult to distinguish from other colors. Where are the theca cells in each time point ? Given the importance of theca cells it would be beneficial to either designate a different color to theca cells or add an additional image of the ovary showing the location of only theca, similar to SF4A, but focusing on theca cells.

2. Only 16 oocytes were identified and sequenced (Fig. 1C). Do the authors know what caused such low yield of oocytes? In the methods section the authors mentioned that they removed cells that had >20% mitochondrial DNA. It is known that oocytes carry a significantly greater number of mitochondria. How did the authors made sure that they did not exclude oocytes by filtering cells with high mitochondria content. 

3. For the explanation of why oocytes were filtered out provided in the discussion (Lines 494-495) "It is likely that the majority of oocytes were filtered out during single-cell preparation suspension, and immature oocytes may have passed through the filter due to their smaller size [118]." do the authors claim that the oocytes were smaller than the somatic cells? What other cells were filtered out in that case? I think an explanation in depth, including sizes that are retained, would sharpen the explanation. 

4. Fig. 1C would it be possible to add the names of the cell types on top of the bar? The colors are quite close, and it would make reading the heat map easier (very minor). 

5. From the description it was not clear how the authors reconciled the differences between the ovaries at the 12h time point in their transcriptomics analysis. Which ovaries (with what phenotype) did they analyze? How do the authors explain such striking differences? It is critical to analyze additional ovaries to determine whether it was a technical mistake, or mice can have two different phenotypes. Was this phenomenon specific to the mouse where both ovaries looked the same or in each mouse one ovary was lagging?

6. With regards to stroma cell analysis did the authors identify any spatial differences between stroma 1 and stroma 2? Were these two cell types localized differently, or the changes only happened transcriptionally? 

7. I am not sure why it wasn't included (technical limitation?), but in general, because of the unique iST, a higher resolution images zoomed in and spatially mapping the important discoveries in this manuscript related to the early and late stages of theca cells and temporal differences in granulosa and luteinized cells it would significantly elevate the depth and contribution of this dataset to the scientific community. However, if it is a technical limitation it should be mentioned in the paper.

8. A similar comment to #7 also goes towards the two types of endothelial cells and their importance during peri- and postovulatory remodeling and the comment about stroma cells and angiogenesis markers and cell-cell interactions of Ang-Egfr.

---

## [Editor Report · Decision Letter 2]

24 Mar 2025

Dear Dr Goods,

Thank you for your patience while we considered your revised manuscript "Single-cell and spatiotemporal profile of ovulation in the mouse ovary" for publication as a Methods and Resources article at PLOS Biology. This revised version of your manuscript has been evaluated by the PLOS Biology editors and the Academic Editor, who is satisfied by your responses to reviewers. 

Based on our Academic Editor's assessment of your revision, we are likely to accept this manuscript for publication. However, before we can editorially accept your study we need you to address a number of data and other policy-related requests, detailed below. 

**IMPORTANT: Please address the following editorial requests: 

1) FINANCIAL DISCLOSURES: Thank you for including a detailed financial disclosures statement which for the most part looks good. However, to be fully compliant, we ask that you update the last sentence of the statement to explicitly detail whether the funders played a role in the study design, data collection and analysis, decision to publish, or preparation of the manuscript (right now, I see the statement says "None of the funders listed above were involved in the design and execution of this study" but we need you to also include a note about whether they were involved in the other aspects, detailed above). 

2) DATA: I see that currently your data availability statement says "Cell-level data and gene-level data will be made available at the time of final publication."

>>Please do make all the underlying data (including the raw scRNA-seq and merfish data) publicly available via a data repository such as Gene Expression Omnibus, Zenodo, etc. 

>>Once that is done, please also ensure that figure legends in your manuscript (including supplemental) include a sentence to direct readers to where the underlying data can be found. Please also update the data availability statement with these details.

For more details on our data policy, which requires that all data be made available without restriction, see: http://journals.plos.org/plosbiology/s/data-availability

or this editorial: http://dx.doi.org/10.1371/journal.pbio.1001797

3) CODE: I see that your paper currently states "All code used in this manuscript for data processing and analysis will be made available on GitHub prior to final publication." 

>>Please do provide any custom code that was generated during the course of this investigation, as this will be required for publication. Please ensure that the code is sufficiently well documented and reusable, and that your Data Statement in the Editorial Manager submission system accurately describes where your code can be found. 

>>Note: we cannot accept sole deposition of code in GitHub, as this could be changed after publication. However, you can archive this version of your publicly available GitHub code to Zenodo. Once you do this, it will generate a DOI number, which you will need to provide in the Data Accessibility Statement (you are welcome to also provide the GitHub access information). See the process for doing this here: https://docs.github.com/en/repositories/archiving-a-github-repository/referencing-and-citing-content

We expect to receive your revised manuscript within two weeks. 

*Published Peer Review History*

*Press*

Sincerely,

Luke

Lucas Smith, Ph.D.

Senior Editor

lsmith@plos.org

PLOS Biology

---

## [Editor Report · Decision Letter 3]

1 May 2025

Dear Britt,

Thank you for the submission of your revised Methods and Resource article "Single-cell and spatiotemporal profile of ovulation in the mouse ovary" for publication in PLOS Biology and thank you for addressing our editorial requests in this revision. On behalf of my colleagues and the Academic Editor, Masahito Ikawa, I am pleased to say that we can in principle accept your manuscript for publication, provided you address any remaining formatting and reporting issues. These will be detailed in an email you should receive within 2-3 business days from our colleagues in the journal operations team; no action is required from you until then. Please note that we will not be able to formally accept your manuscript and schedule it for publication until you have completed any requested changes.

**IMPORTANT: Please note that I have updated your 'data availability statement' to include the new DOI that you provided me for your spatial omics dataset. Please take a moment to double check that everything looks good after this change. 

PRESS

Sincerely, 

Luke 

Lucas Smith, Ph.D.

Senior Editor

PLOS Biology

lsmith@plos.org